# Transformers are adaptable task planners

**Vidhi Jain**[1,2]**, Yixin Lin**[1]**, Eric Undersander**[1]**, Yonatan Bisk**[2]**, Akshara Rai**[1]
[1] Meta, [2] Carnegie Mellon University

**Abstract:** Every home is different, and every person likes things done in their own way. Therefore, home robots of the future need to both reason about the sequential nature of day-to-day tasks and generalize to user's preferences. To this end, we propose a Transformer Task Planner (TTP) that learns high-level reasoning from demonstrations by leveraging object attribute-based representations. TTP is pre-trained on multiple preferences in a simulated dishwasher loading task and shows generalization to *unseen* preferences using a single demonstration as a prompt. Further, we demonstrate real-world dish rearrangement using TTP with a Franka robotic arm. See videos and code at this website.

**Keywords:** Task Planning, Prompt, Preferences, Object-centric Representation

## 1 Introduction

Consider a robot tasked with loading a dishwasher. Such a robot has to account for task constraints (e.g. only an open dishwasher rack can be loaded), and dynamic environments (e.g. more dishes may arrive once the robot starts loading the dishwasher). Dishwasher loading is such a canonical task with user-specific preferences, like a user may place mugs on the top rack and plates on the bottom or load dirtier bowls before easier-to-clean cups, encoding their preference. Additionally, they pull out a rack before loading it, inherently encoding a structural constraint. This also holds in other household tasks, like cooking. For example, a person may start cooking potatoes before frying onions, as potatoes often take longer to cook, while another person first caramelizes the onions before adding in potatoes for flavor. In a kitchen, a user might have a preferred cabinet for plates, and another for cups. Learning temporal and spatial constraints and preferences from demonstrations requires policies with temporal context that can consider the *sequence of actions* demonstrated. Classical task planning [1] deals with such constraints through symbolic task description, but such descriptions are difficult to design and modify for new preferences in complex tasks. Building easily adaptable long-horizon task plans, under constraints and uncertainty, is an open problem in robotics.

Transformers are well-suited to this problem, as they have been shown to learn long-range relationships [2], although not in temporal robotic tasks. Recent work [3] has shown Transformers [4] can learn temporally-consistent representations, and generalize to new scenarios [5, 6, 7]. Our central question is: *Can a Transformer learn task structure, adapt to user preferences, and achieve complex long-horizon tasks using no symbolic task representations?* We propose Transformer Task Planner (TTP) - an adaptation of a classic transformer architecture that includes temporal, pose, and category embeddings to learn object-oriented relationships over space and time. By pre-training TTP on multiple preferences, TTP learns to infer unseen preferences from a single demonstration and adhere to it with a variable number of objects and dynamic environments.

Our main contributions are to (i) introduce transformers as a promising architecture for learning task plans from demonstrations using object-centric embeddings (Sec. 2), (ii) show that preference-conditioned pre-training generalizes at test time to new, unseen preferences and outperforms competitive baselines (Sec. 3.1), and (iii) transfer TTP to a rearrangement problem in the real world, where a Franka arm places dishes in two drawers, using a single demonstration (Fig. 1 and Sec. 3.2).

6th Conference on Robot Learning (CoRL 2022), Auckland, New Zealand.

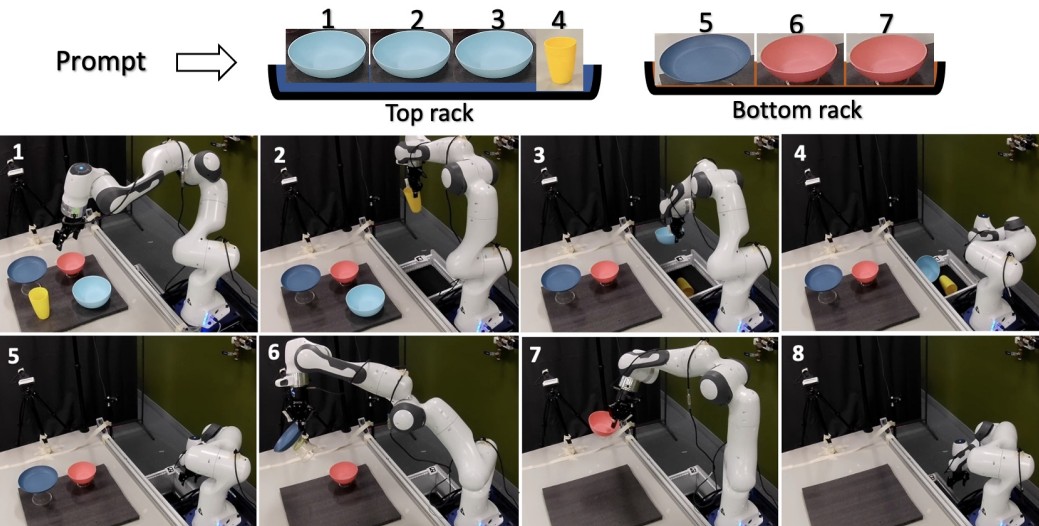

Figure 1: TTP makes decisions about "when to open drawers" and "what objects to pick" for a robot arm, such that it follows the preferred order and location, as indicated in a prompt demonstration. **Top**: Prompt summarized as an illustration. **Bottom**: Robot loads 4 dishes - (1-4) opens top drawer and places blue bowl and cup, then (5-8) opens bottom drawer to place plate and pink bowl. Note that the prompt can have different number of objects than in the situation where the robot acts. TTP learns to infer and apply preferences in simulation and transfers zero-shot to the real world.

## 2 Transformer Task Planner (TTP)

We introduce TTP, a transformer-based policy architecture for learning sequential manipulation tasks. We assume low-level generalized pick-place primitive actions that apply to both objects like plates and bowls, as well as the dishwasher door and racks. TTP learns a high-level policy for pick-place in accordance with the task structure and preference shown in demonstrations. The following sections are described with dishwasher-loading as an example, but our setup is applicable to most long-horizon manipulation tasks with generalized pick-place.

**State-Action Representations** We consider a high-level policy that interacts with the environment in discrete time steps. At every timestep $t$, we receive observation $\boldsymbol{o}_t$ from the environment which passes through a perception pipeline to produce a set of rigid-body 'pick'-able instances $\{x_i\}_{i=1}^n$, corresponding to the $n$ objects currently visible. The pick *state* $S_t^{pick}$ is described as the set of instances that are **visible** in $\boldsymbol{o}_t$ : $S_t^{pick} = \{x_1, x_2, \cdots, x_n\}$. Once a 'pick' object is chosen, we decide where to place it. We pre-compile a list of discrete placement poses corresponding to viable place locations for each object category, created by randomly placing objects in the dishwasher and measuring the final pose they land in. All possible placement locations for the picked object category, whether free or occupied, are used to create a set of 'place' instances $\{g_j\}_{j=1}^l$. We describe an *action* $\boldsymbol{a}_t$ in terms of what to pick and where to place it. Specifically, a policy $\pi$ outputs the pick action, which is an object in $\boldsymbol{o}_t$, i.e. $\pi(S^{pick}) \to x_{target}$. The place state is $S^{place} = S^{pick} \cup \{g_j\}_{j=1}^l$. The *same* policy $\pi$ then chooses where to place the object, $\pi(S^{place}) \to g_{target}$ (Fig. 2). Note that the input for predicting place includes both objects and place instances since the objects determine whether a place instance is free to place or not. On the other hand, the 'pick' action only looks at the current object instances to determine the next best object to pick. A demonstration is a state-action sequence $\mathcal{C} = \{(S_0, a_0), (S_1, a_1), \cdots, (S_{T-1}, a_{T-1}), (S_T)\}$. See B.1.

**Instance Encoder** We describe both 'pick' and 'place' instances in terms of their attributes such as $\{p, c, t, r\}$, where $p$ is the pose of the object, $c$ is its category, $t$ is the timestep while recording $\boldsymbol{o}_t$ and $r$ is a boolean indicator, for instance, type – 'pick' or 'place' (Fig. 2a). The pose is embedded as $\Phi_p$, using a positional encoding scheme similar to NeRF [8] to encode the 3D positional coordinates and 4D quaternion rotation. To encode the category, we use the dimensions of the 3D bounding box

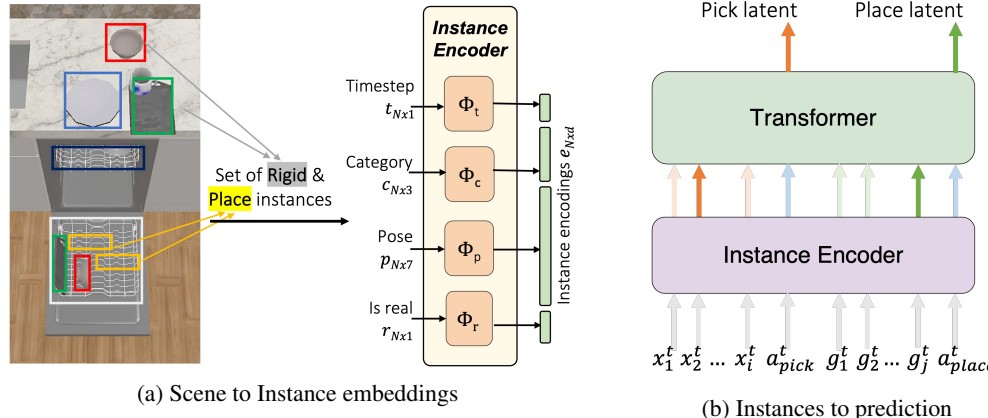

Pick latent      Place latent

(a) Scene to Instance embeddings

(b) Instances to prediction

Figure 2: (Left) Architecture overview of how a scene is converted to a set of instances. Each instance is comprised of attributes, i.e. pose, category, timestep, and whether it is an object or place instance. (Right) Instance attributes (gray) are passed to the encoder, which returns instance embeddings for pickable objects (red), placeable locations (green), and <ACT> embeddings (blue). The transformer outputs a chosen pick (red) and place (green) instance embedding.

of the object to build a continuous space of object types and process this through an MLP $\Phi_c$. For each discrete 1D timestep, we model $\Phi_t$ as a learnable embedding in a lookup table, similar to the positional encodings in BERT [9]. The concatenated $d$-dimensional embedding for an instance at timestep $t$ is represented as $f_e(x_i) = \Phi_t || \Phi_c || \Phi_p || \Phi_r = e_i$. See E.1 for instance encoding design. The encoded state at time $t$ is represented as: $S_t^{enc} = \{e_0, e_1, \cdots e_N\}$. We drop $()^{enc}$ for brevity.

## 2.1 Prompt-Situation Transformer

We use Transformers [4], a deep neural network that operates on sequential data, to learn a high-level pick-place policy $\pi$. The input to the encoder is a $d$-dimensional token[1] per instance $e_i \in [1, ..N]$ in the state $S$. In addition to instances, we introduce special <ACT> tokens[2], a zero vector for all attributes, to demarcate the end of one state and start of the next. These tokens help maintain the temporal structure; all instances between two <ACT> tokens in a sequence represent one observed state. A demonstration without actions is $\tau = [S_{t=0}, <\text{ACT}>, \cdots, S_{t=T-1}, <\text{ACT}>, S_{t=T}]$. To learn a common policy for multiple preferences, we propose a prompt-situation architecture, where "prompt" is a demonstration with some underlying preference, and "situation" is a different environment scenario where the policy acts. The policy cannot simply copy the prompt actions because the prompt object set is different from the situation. Instead, it needs to infer the underlying preference from the prompt and then apply it in the given situation.

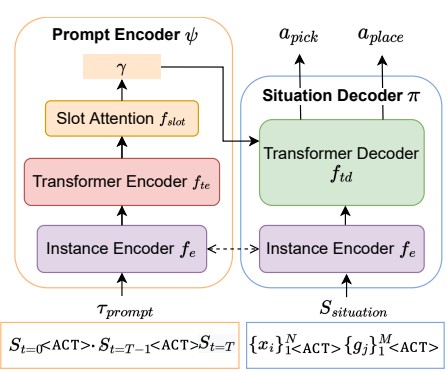

Figure 3: Prompt-Situation Architecture. Left-half: prompt encoder $\psi$ that takes as input a "prompt" (demonstration) and outputs a learned preference embedding $\gamma$. Right-half: the situation decoder $\pi$, which is conditioned on preference embedding from prompt encoder and is acting on the state from given "situation" (environment).

In Fig. 3, the prompt encoder $\psi$ is on the left and the situation decoder or policy $\pi$ acting on the given state of the situation is on the right. The prompt encoder $\psi : f_{slot} \circ f_{te} \circ f_e$ consists of an instance encoder $f_e$, transformer encoder $f_{te}$, and a slot-attention layer $f_{slot}$ [10]. Slot attention

[1]Terminology borrowed from natural language processing where tokens are words; here, they are instances.
[2]Similar to $<CLS>$ tokens used for sentence classification.

is an information bottleneck, which learns semantically meaningful representations of the prompt as a fixed and reduced dimensional embedding. $\psi$ takes the demonstration $\tau_{\text{prompt}}$ as input and returns $\gamma = \psi(\tau_{\text{prompt}})$. The situation decoder $\pi : f_{td} \circ f_e$ receives as input $N$ instance tokens from the current scene $S$, consisting of objects, as well as, placement instances (Fig. 2b). It consists a transformer decoder [4] with self-attention layers on the $N$ input tokens, followed by a cross-attention layer with preference embedding $\gamma$ from the prompt encoder. We select the output of the situation decoder at the $<$ACT$>$ token and calculate dot-product similarity with the $N$ input tokens $e_i$. The token with the maximum dot-product is chosen as the predicted instance: $x_{pred} = \max_{e_i, i \in \{1, N\}} \left( \hat{x}_{<\text{ACT}>} \cdot e_i \right)$. $\pi$ is trained with cross-entropy to maximize the similarity of output latent with the expert's chosen instance embedding as target. To summarize, the prompt encoder receives a prompt demonstration as input and outputs a learned representation of the preference. These output prompt tokens are input to a situation decoder $\pi$, which also receives the current state as input. The decoder $\pi$ predicts the action chosen by the expert in the given situation, adhering to the preference indicated in the prompt.

## 2.2 Multi-Preference Task Learning

We adopt a prompt-situation architecture for multi-preference learning. This design (1) enables multi-preference training by disambiguating preferences, (2) learns task-level rules shared between preferences (e.g. dishwasher should be open before placing objects), (3) can generalize to unseen preferences at test time, without fine-tuning. Given a 'prompt' demonstration of preference $m$, our policy semantically imitates it in a different 'situation' (i.e. a different initialization of the scene). To this end, we learn a representation of the prompt $\gamma^m$ which conditions $\pi$ to imitate the expert.

$$\psi(\tau_{\text{prompt}}^m) \rightarrow \gamma^m \tag{1}$$

$$\pi(S_{\text{situation}} | \gamma^m) \rightarrow a_t^m = \{x_{\text{pred}}^m, g_{\text{pred}}^m\} \tag{2}$$

We train neural networks $\psi$ and $\pi$ together to minimize the total prediction loss on all preferences using the multi-preference training dataset $\mathcal{D}$. The overall objective is:

$$\min_{m \sim M, \tau \sim \mathcal{D}_m, (S,a) \sim \mathcal{D}_m} \mathcal{L}_{CE}(a, \pi(S | \psi(\tau))) \tag{3}$$

For every preference $m$ in the dataset, we sample a demonstration from $\mathcal{D}_m$ and use it without actions as a prompt. We also sample a state-action pair $(S, a)$ from any demonstration $\mathcal{C} \in \mathcal{D}_m$. At test time, we record one prompt demo from a seen or unseen preference to output action: $a = \pi(S | \psi(\tau_{\text{prompt}}))$. All policy weights are kept fixed during testing, and generalization to new preferences is zero-shot using the learned preference representation $\gamma$. Unlike [11], $\gamma$ captures not just the final state, but a temporal representation of the whole demonstration. Building a temporal representation is crucial to encode demonstration preferences like the order of loading racks and objects. Even though the final state is the same for two preferences that only differ in which rack is loaded first, our approach is able to distinguish between them using the temporal information in $\tau_{\text{prompt}}$. To the best of our knowledge, our approach is the first to temporally encode preferences inferred from a demonstration in learned task planners.

## 3 Experiments

We present the "Replica Synthetic Apartment 0 Kitchen"[3] (see figure 4, appendix and video), an artist-authored interactive recreation of the kitchen of the "Apartment 0" space from the Replica dataset [12]. We use selected objects from the ReplicaCAD [13] dataset, including seven types of dishes, and procedurally generate dishwasher loading demonstrations. See B.1.

---

[3]"Replica Synthetic Apartment 0 Kitchen" was created with the consent of and compensation to artists and will be shared under a Creative Commons license for non-commercial use with attribution (CC-BY-NC).

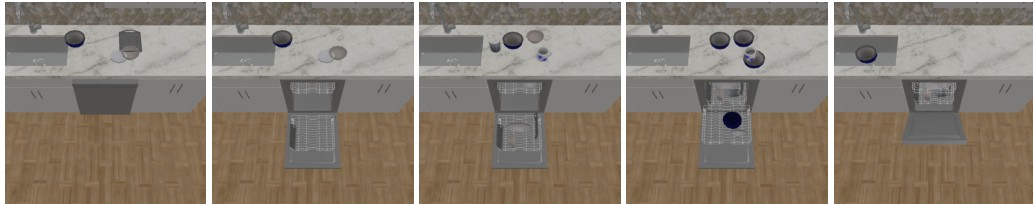

Figure 4: Dishwasher Loading demonstration in AI Habitat Kitchen Arrange Simulator. Objects dynamically appear on the countertop (ii-iv), and need to be placed in the dishwasher. If the dishwasher racks are full, they land in sink (v).

We define a preference in terms of expert demonstration 'properties', like which rack is loaded first with what objects? Table 1 describes the preferences of dishwasher loading in terms of three properties - first loaded tray, objects in the top and bottom tray. Preferences 1 & 2 vary in the order of which rack is loaded first, while 2 & 3 both load the bottom rack first with similar categories on top and bottom but with different orderings for these categories. Our dataset consists of 12 preferences (7 train, 5 held-out test) with 100 sessions per preference. In a session, $n \in \{3, ..., 10\}$ instances are loaded in each rack. The training data consists of sessions with 6 or 7 objects allowed per rack. The held-out test set contains 5 unseen preferences for $\{3, 4, 5, 8, 9, 10\}$ objects per rack. See B.2. Additionally, to simulate a dynamic environment, we randomly initialize new objects mid-session on the kitchen counter. This simulates situations where the policy does not know every object to be loaded at the start of the session and has to learn to be reactive to new information. See B.3.

Table 1: Three example preferences for dishwasher loading. Rack order and their respective contents (ordered by preference).

| First? | Top | Bottom |
|---|---|---|
| Top | 1. cups
2. glasses
3. small bowl | 1. big plates
2. small plates
3. trays
4. big bowls |
| Bottom | 1. cups
2. glasses
3. small bowl | 1. big plates
2. small plates
3. trays
4. big bowl |
| Bottom | 1. small plate
2. glasses
3. cups | 1. big bowls
2. trays
3. big plates
4. small bowl |

To measure success, we rollout the policy from an initial state and compare the resultant trajectory with an expert demonstration. Rollouts require repeated decisions in the environment, without any resets. A mistake made early on in a rollout can result in poor performance, even if the prediction accuracy is high. For example, if a policy mistakenly fails to open a dishwasher rack, the performance will be poor, despite good prediction accuracy in later steps. For in-distribution evaluation, 10 sessions are held-out per preference. For out-of-distribution evaluation, we create sessions with unseen preferences and unseen numbers of objects. Specifically, we measure:

**Spatial Preference Adherence (SPA)** : SPA measures how well is the final state packed per user's spatial preference. Let $\hat{a}_i$ be the number of objects placed in the $i^{th}$ receptacle adhering to the preference by the learned policy, and $a_i$ be the number for an expert, then $SPA = \frac{1}{N} \sum_{i=1}^{N} \frac{\hat{a}_i}{max(a_i, \hat{a}_i)}$. SPA measures the packing efficiency of the final state of the dishwasher after policy rollout, while respecting the spatial preference of top or bottom rack per category. Note that if the policy follows the wrong spatial preference (places objects in the wrong rack), then SPA is low, even if the dishwasher is full.

**Temporal Preference Adherence (TPA)**: TPA measures how much the policy deviates from the order of actions shown in an expert demonstration. Let $\hat{\tau}, \tau$ be the sequence of picked instances by the learned policy and expert respectively, then TPA $= 1 - \frac{EditDistance(\hat{\tau}, \tau)}{max(|\tau|, |\hat{\tau}|)}$. TPA accounts for the order in which objects are picked, irrespective of where they are placed, and measures the deviation from the temporal actions of the user. TPA is high if the policy follows both sequential constraints of task (open dishwasher before loading), and the order-related preferences (bowls before cups).

If the expert preference was to load the top rack first and the learned policy loads the bottom first, SPA would be perfect at 1.0, but TPA would be low. Both metrics are low if the policy repeatedly violates the task's physical constraints or does not make progress towards the task. See C.4.

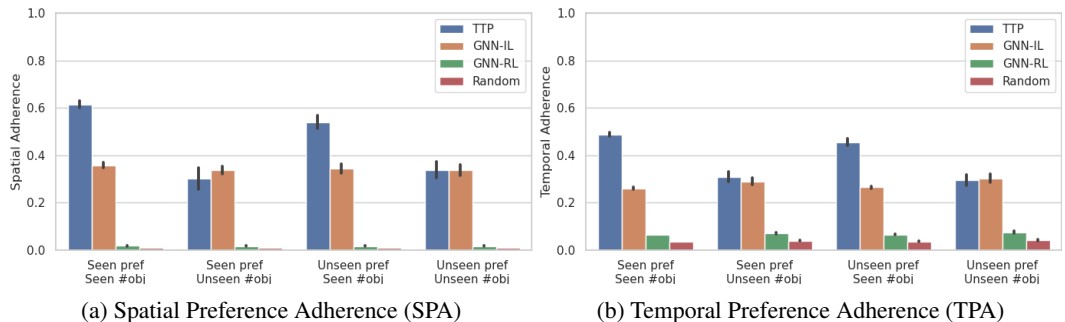

(a) Spatial Preference Adherence (SPA)   (b) Temporal Preference Adherence (TPA)

Figure 5: Comparisons of TTP, GNN-IL/RL and a Random policy in simulation across two metrics: SPA and TPA. TTP shows good performance at the task of dishwasher loading on seen and unseen preferences, and outperforms the GNN and random baselines. However TTP's generalization to unseen # objects is worse, but still close to GNN-IL (which was trained on *all* preferences), and better than GNN-RL and random.

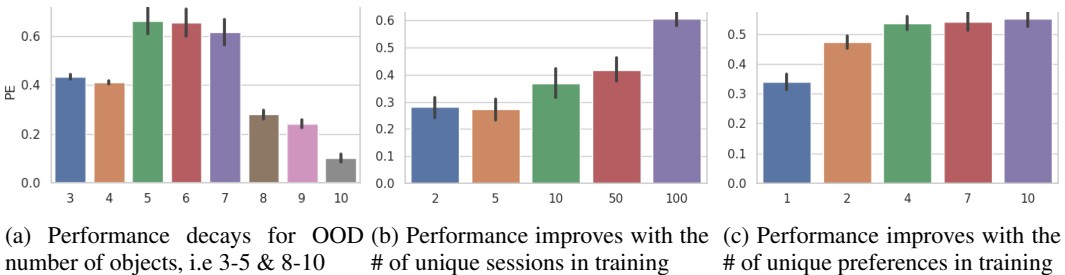

(a) Performance decays for OOD number of objects, i.e 3-5 & 8-10 | (b) Performance improves with the # of unique sessions in training | (c) Performance improves with the # of unique preferences in training

Figure 6: TTP experiments (a) Out-of-distribution generalization to #objects. (b-c) ablations.

## 3.1 Simulated Dishwasher Loading

We compare our approach against Graph Neural Network (GNN) based preference learning from [14, 11] at this task of dishwasher loading. Neither of these works is directly suitable for our task, so we combine [14, 11] to create a stronger baseline. We use ground-truth preference labels, represented as a categorical distribution, and add them to the input features of the GNN, similar to [11]. We use the output of the GNN to make sequential predictions, as in [14]. Thus, by combining the two works, we create a GNN baseline that can act according to preference in a dishwasher-loading scenario. The GNN policy is trained on all preferences, including the 'unseen' preferences. **For unseen preferences, this is privileged information** that our approach does not have access to. We train the GNN policy using imitation learning (IL), and reinforcement learning (RL), following [14]. GNN-IL is trained from the same set of demonstrations as TTP using behavior cloning (see Fig. 12a). For GNN-RL, we use Proximal Policy Optimization [15] from [16]. GNN-RL learns from scratch by directly interacting with the dishwasher environment and obtaining a dense reward. For more details on baselines, including an alternate variant of GNN-IL, see Appendix C.

GNN-IL does not reach the same performance as TTP for in-distribution tasks (SPA of $0.34$ for GNN-IL vs $0.62$ for TTP). Note that unseen preferences are also in-distribution for GNN-IL since we provide ground-truth preference labels to the GNN. Hence, there isn't a drop in performance for unseen preferences for GNN, unlike TTP. For detailed failure analysis, see Appendix D. Despite having no privileged information, TTP outperforms GNN-IL in unseen preferences and performs comparably on unseen #objects. Due to the significantly long time horizons per session (more than 30 steps), GNN-RL fails to learn a meaningful policy even after a large budget of 32,000 environment interactions and a dense reward (SPA $0.017 \pm 0.002$ using GNN-RL). Lastly, we find that the random policy RP is not able to solve the task at all due to the large state and action space. TTP is able to solve dishwasher loading using unseen preference prompts well (SPA $0.54$). In contrast, classical task planners like [1] need to be adapted per new preference. This experiment shows that Transformers make adaptable task planners, using our proposed prompt-situation architecture.

However, TTP's performance on unseen #objects deteriorates (SPA of 0.62 for seen versus 0.34 on unseen #objects) , and we look more closely at that next.

**Generalization to unseen # objects**: Fig.6a examines SPA on out-of-distribution sessions with lesser i.e. 3-5 or more i.e. 8-10 objects per rack. The training set consists of demonstrations with 6 or 7 objects per rack. The policy performs well on 5-7 objects, but poorer as we go further away from the training distribution. Poorer SPA is expected for larger number of objects, as the action space of the policy increases, and the policy is more likely to pick the wrong object type for a given preference. Poorer performance on 3-4 objects is caused by the policy closing the dishwasher early, as it has never seen this state during training. Training with richer datasets, and adding more randomization in the form of object masking might improve out-of-distribution performance of TTP.

### 3.1.1  Ablations

We study the sensitivity of our approach to training hyperparameters. First, we vary the number of training sessions per preference and study generalization to unseen scenarios of the same preference. Figure 6b shows that the performance of TTP improves as the number of demonstrations increases, indicating that our model might benefit from further training samples. Next, we vary the number of training preferences and evaluate generalization performance to unseen preferences. Figure 6c shows that the benefits of adding additional preferences beyond 4 are minor, and similar performance is observed when training from 4-7 preferences. This is an interesting result since one would assume that more preferences improve

Table 2: Attribute Ablations

| Pose | Cat | Time | SPA |
|------|-----|------|-------|
| × | × | × | 0.0 |
| × | × | ✓ | 0.0 |
| × | ✓ | × | 0.027 |
| × | ✓ | ✓ | 0.142 |
| ✓ | × | × | 0.411 |
| ✓ | × | ✓ | 0.419 |
| ✓ | ✓ | × | 0.517 |
| ✓ | ✓ | ✓ | 0.606 |

generalization to unseen preferences. But for the kinds of preferences considered in our problem, 4-7 preference types are enough for generalization.

Finally, we analyze which instance attributes are the most important for learning in an object-centric sequential decision-making task. We mask different instance attributes to remove sources of information from the instance tokens. In Table 2, all components of our instance tokens play a significant role (0.606 with all, versus the next highest of 0.517). The most important attribute is the pose of the objects (without the pose, top SPA is 0.142), followed by the category. The timestep is the least important, but the best SPA comes from combining all three. More ablations are in Appendix E.

### 3.2  Real-world dish-rearrangement

As a proof-of-concept, we demonstrate zero-shot transfer of our policy trained in simulation to robotic hardware, by assuming low-level controllers. We use a Franka Panda with a Robotiq 2F-85 gripper, controlled using the Polymetis control framework [17]. For perception, we use three Intel Realsense D435 RGBD cameras [18] whose outputs are combined using Open3D [19] and segmented [20] to generate 3D point clouds of visible objects, used to calculate object-centric attributes used by TTP. For low-level pick, we use a grasp candidate generator [21] applied to a chosen object's point cloud and use it to grasp the target object. 'Place' is approximated as a 'drop' action in a pre-defined location. More details are in Appendix A.

Our hardware setup mirrors our simulation, with different categories of dishware (bowls, cups, plates) on a table, and a "dishwasher" (cabinet with two drawers). The objective is to select an object to pick and place it into a drawer (rack) (see Fig. 1). We use a policy trained in simulation and apply it to a scene with four objects (2 bowls, 1 cup, 1 plate) using the hardware pipeline described above. We start by collecting a prompt human demonstration which is manually segmented into pick and place motions, and the environment state recorded at each timestep. The learned policy, conditioned on the prompt, is applied to two variations of the scene, and the predicted actions executed. The hardware success is defined in terms of pick only, shown in Fig. 1, since we do not reason about 'place' on hardware. The policy successful places all 4 objects in the correct order in the drawer once and 3 out of 4 objects in the second attempt. The hardware failure case was

caused by a perception error; a bowl was classified as a cup. This demonstrates that TTP can be trained in simulation and applied directly to hardware, and the policy is robust to minor hardware errors, like noise in object location. It is also reactive to failure, such as if a bowl grasp fails, it just repeats the grasping action. However, it is sensitive to perception errors like failing to detect objects, or mis-classifying them. See Appendix F. In the future, TTP should be evaluated in more diverse real-world settings and its sensitivity to the different hardware components measure more carefully.

## 4  Prior Work

**Learning for Sequential manipulation** aims to overcome the challenges in designing and scaling classical Task and Motion Planning approaches. Object-centric pick-place representations using off-the-shelf perception methods are common in learning policies [22, 23, 24, 25, 26, 27], specifically for sequential manipulation [1, 28]. TransporterNets [29] primarily deal with table-top Pick&Place generalized to new object locations and do not consider in 3D sequential reasoning problems. SOR-Net [30] assumes access to a task planner to reason about sequential reasoning instead of a learned policy. Such a task planner is often tedious to design for complex tasks, and difficult to modify per preference. While [14] presents a learned sequential reasoning approach using GNNs, and [11] shows generalization of GNNs to preferences, neither is directly suitable for our task, so we combine [14, 11] to create a stronger baseline. Recent works have repurposed Transformers for other sequence modeling tasks [31, 32, 33, 34, 35, 36]. Prompt DT [37] considers a single model to encode the prompt and the successive sequence of state. This limits the size of the prompt and requires special tokens to demarcate between prompt and current state. We use the encoder-decoder setup and share the parameters for instance encoding. PlaTe [32] proposes planning from videos, while [34, 35, 36] model a sequential decision-making task. We consider long-horizon tasks with partially observable state features and user-specific preferences.

**Preferences and prompt training** There are several ways of encoding preferences in recent works. [11] propose VAE to learn user preferences for a spatial arrangement based on just the final state, while our approach models temporal preference from demonstrations. Preference-based RL learns rewards based on human preferences [38, 39, 40, 41], but do not generalize to unseen preferences. For complex long-horizon tasks, [42] shows that modeling human preferences enables faster learning than RL. We show generalization to an unseen preference by learning to infer them from the demonstration. Large language and vision models have shown generalization through prompting [43, 44]. Prompting has been used to guide a model to quickly switch between multiple task objectives [45, 46, 47]. Recent language models learn representations that can be easily transferred to new tasks in a few-shot setting [48, 49, 5, 43]. Our approach similarly utilizes prompts for preference generalization in sequential decision-making robotic tasks.

## 5  Conclusions and Limitations

We present Transformer Task Planner (TTP): a high-level, sequential, preference-based policy from a single demonstration using a prompt-situation architecture. We introduced a simulated dishwasher loading task with demonstrations that exhibit different preferences. TTP can solve this complex, long-horizon task in simulation and transfers to the real world. This is a first step towards learning to adapt to different users' preferences in terms of both ordering and spatial location.

While our task is complex by virtue of its strict sequential nature, it is incomplete as we assume doors and drawers can be easily opened and perception is perfect. In complex real settings, the policy needs to learn how to recover from its low-level execution mistakes. Moreover, preferences can have further nuanced differences based on visual or textural patterns on objects, for which the instance encoder needs to be modified to encode such attributes. Another important question to address is how more complex motion plans interact with or hinder the learning objective, due to different human and robot affordances. Finally, prompts are only presented via single demonstration, while language might be a more natural interface for users to instruct the robot.

**Acknowledgments**

We thank Tingfan Wu and Austin Wang for their support in the hardware setup, as well as Amy Zhang, Chris Paxton, Ben Newman, Adriana Romero Soriano, Vikash Kumar, and Dhruv Batra for the helpful initial discussions. We also thank all anonymous reviewers for CoRL'22 and Rishi Veerapaneni for high-quality feedback and suggestions.

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

# A Hardware Experiments

## A.1 Real-world prompt demonstration

Here we describe how we collected and processed a visual, human demonstration in the real-world to treat as a prompt for the trained TTP policy (Fig. 7). Essentially, we collect demonstration pointcloud sequences and manually segment them into different pick-place segments, followed by extracting object states. At each high-level step, we measure the state using three RealSense RGBD cameras[18], which are calibrated to the robot frame of reference using ARTags [50]. The camera output, extrinsics, and intrinsics are combined using Open3D [19] to generate a combined point-cloud. This pointcloud is segmented and clustered to give objects' pose and category using the algorithm from [20] and DBScan. For each object point cloud cluster, we identify the object pose based on the mean of the point cloud. For category information we use median RGB value of the pointcloud, and map it to apriori known set of objects. In the future this can be replaced by more advanced techniques like MaskRCNN [51]. Placement poses are approximated as a fixed, known location, as the place action on hardware is a fixed 'drop' position and orientation. The per step state of the objects is used to create the input prompt tokens used to condition the policy rollout in the real-world, as described in Section 3.2.

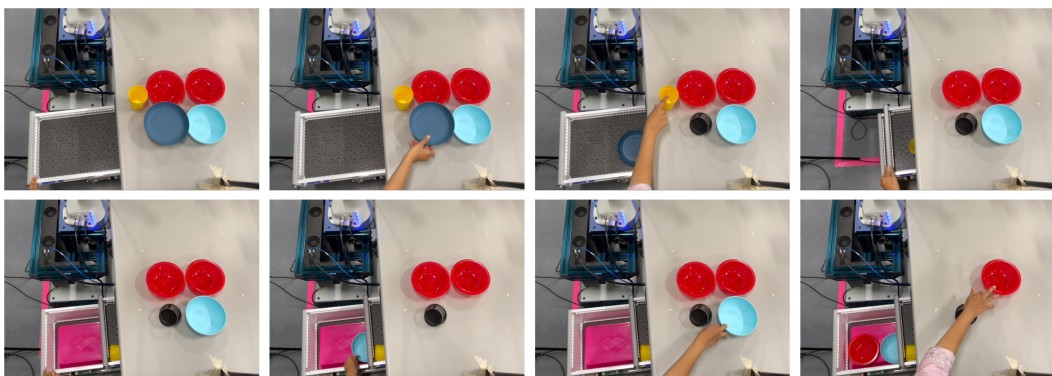

Figure 7: Human demonstration of real-world rearrangement of household dishes.

## A.2 Hardware policy rollout

We zero-shot transfer our policy $\pi$ trained in simulation to robotic hardware, by assuming low-level controllers. We use a Franka Panda equipped with a Robotiq 2F-85 gripper, controlled using the Polymetis control framework [17]. Our hardware setup mirrors our simulation, with different categories of dishware (bowls, cups, plates) on a table, a "dishwasher" (cabinet with two drawers). The objective is to select an object to pick and place it into a drawer (rack) (see Fig. 7).

Once we collect the human prompt demonstration tokens, we can use them to condition the learned policy $\pi$ from simulation. Converting the hardware state to tokens input to $\pi$ follows the same pipeline as the ones used for collecting human demonstrations. At each step, the scene is captured using 3 Realsense cameras, and the combined pointcound is segmented and clustered to get object poses and categories. This information along with the timestep is used to generate instance tokens as described in Section 2 for all objects visible to the cameras. For visible already placed objects, the place pose is approximated as a fixed location. The policy $\pi$, conditioned on the human demo, reasons about the state of the environment, and chooses which object to pick. Next, we use a grasp generator from [21] that operates on point clouds to generate candidate grasp locations on the chosen object. We filter out grasp locations that are kinematically not reachable by the robot, as well as grasp locations located on points that intersect with other objects in the scene. Next, we select the top 5 most confident grasps, as estimated by the grasp generator, and choose the most top-down grasp. We design an pre-grasp approach pose for the robot which is the same final orientation as

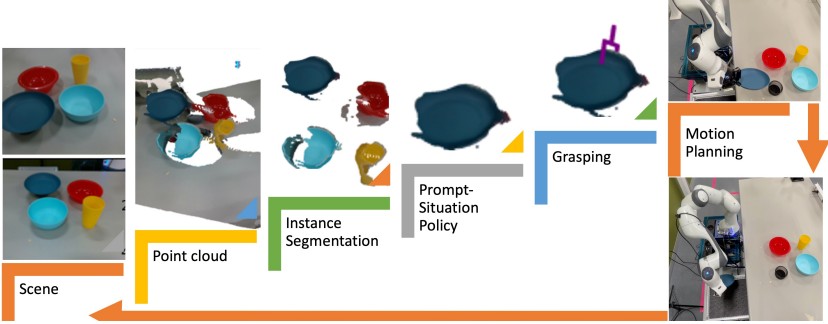

Figure 8: Pipeline for Real Hardware Experiments

the grasp, located higher on the grasping plane. The robot moves to the approach pose following a minimum-jerk trajectory, and then follows a straight line path along the approach axes to grasp the object. Once grasped, the object is moved to the pre-defined place pose and dropped in a drawer. The primitives for opening and closing the drawers are manually designed on hardware.

The learned policy, conditioned on prompt demonstrations, is applied to two variations of the same scene, and the predicted pick actions are executed. Fig.9 shows the captured image from one of the three cameras, the merged point cloud and the chosen object to pick and selected grasp for the same. The policy was successful once with 100% success rate, and once with 75%, shown in Fig.1. The failure case was caused due to a perception error – a bowl was classified as a plate. This demonstrates that our approach (TTP) can be trained in simulation and applied directly to hardware. The policy is robust to minor hardware errors like a failed grasp; it just measures the new state of the environment and chooses the next object to grasp. For example, if the robot fails to grasp a bowl, and slightly shifts the bowl, the cameras measure the new pose of the bowl, which is sent to the policy. However, TTP relies on accurate perception of the state. If an object is incorrectly classified, the policy might choose to pick the wrong object, deviating from the demonstration preference. In the future, we would like to further evaluate our approach on more diverse real-world settings and measure its sensitivity to the different hardware components, informing future choices for learning robust policies.

## A.3  Transforming hardware to simulation data distribution

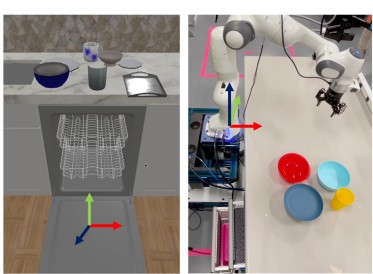

Figure 10: Coordinate Frame of reference in simulation (left) and real world setting (right). Red is x-axis, green is y-axis and blue is z-axis.

The policy trained in simulation applies zero-shot to real-world scenarios, but it requires a coordinate transform. Fig. 10 shows the coordinate frame of reference in simulation and real world setting. Since our instance embedding uses the poses of objects, it is dependant on the coordinate frame that the training data was collected in. Since hardware and simulation are significantly different, this coordinate frame is not the same between sim and real. We build a transformation that converts hardware measured poses to the simulation frame of reference, which is then used to create the instance tokens. This ensures that there is no sim-to-real gap in object positions, reducing the challenges involved in applying such a simulation trained policy to hardware. In this section we describe how we convert the real world coordinates to simulation frame coordinates for running the trained TTP policy on a Franka arm.

We use the semantic work area in simulation and hardware to transform the hardware position coordinates to simulation position coordinates. We measure the extremes of the real workspace by manually moving the robot to record positions and orientations that define the extents of the workspace for the table. The extents of the drawers are measured by placing ARTag markers. We build 3

## RGB images     Point Cloud     Grasps

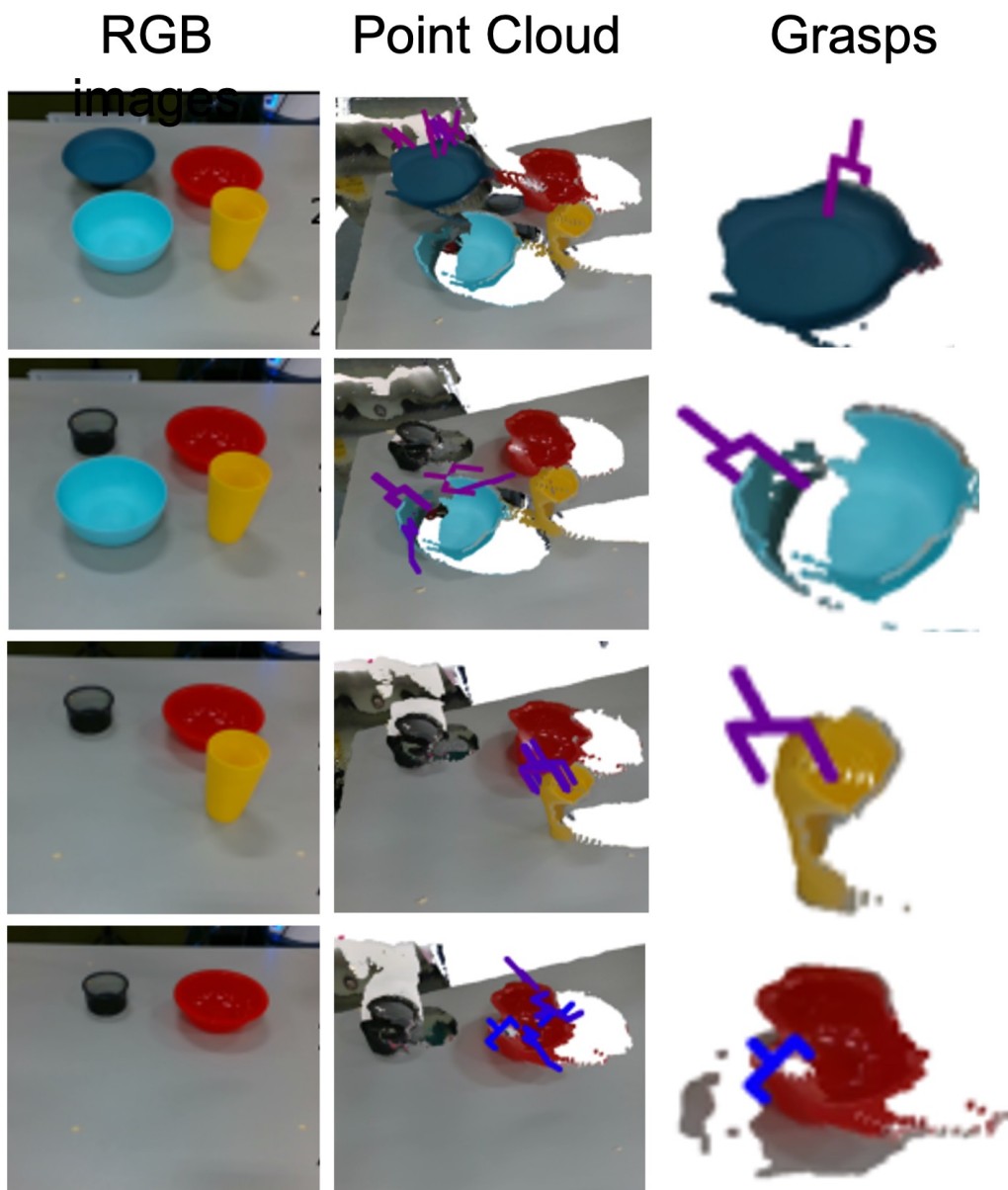

Figure 9: Point cloud and grasps for different objects during policy rollout.

real-to-sim transformations using the extents for counter, top rack and bottom rack: Let $X \in \mathbb{R}^{3 \times N}$ contain homogeneous $xz-$ coordinates of a work area, along its column, as follows:

$$
X = \begin{bmatrix} x^{(1)} & x^{(2)} & \cdots \\ z^{(1)} & z^{(2)} & \cdots \\ 1 & 1 & \cdots \end{bmatrix} = \begin{bmatrix} \boldsymbol{x}^{(1)} & \boldsymbol{x}^{(2)} & \cdots \end{bmatrix} \tag{4}
$$

As the required transformation from real to simulation involves scaling and translation only, we have 4 unknowns, namely, $\boldsymbol{a} = [\alpha_x, \alpha_y, x_{trans}, z_{trans}]$. Here $\alpha_x, \alpha_z$ are scaling factors and $x_{trans}, z_{trans}$ are translation offset for $x$ and $z$ axis respectively. To solve $X_{sim} = AX_{hw}$, we need to find the transformation matrix $A = \hat{\boldsymbol{a}} = \begin{bmatrix} \alpha_x & 0 & x_{trans} \\ 0 & \alpha_z & z_{trans} \\ 0 & 0 & 1 \end{bmatrix}$.

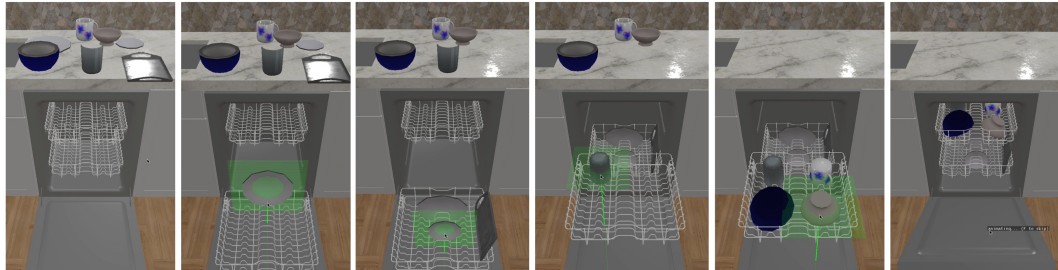

Figure 11: Human demonstration with point and click in simulation

$$X_{sim} = \hat{\boldsymbol{a}} X_{hw} \tag{5}$$

Rewriting the  system of linear equations, (6)

$$\implies \begin{bmatrix} x_{sim}^{(1)} \\ z_{sim}^{(1)} \\ x_{sim}^{(2)} \\ z_{sim}^{(2)} \\ \vdots \end{bmatrix} = \begin{bmatrix} x_{hw}^{(1)} & 0 & 1 & 0 \\ 0 & z_{hw}^{(1)} & 0 & 1 \\ x_{hw}^{(2)} & 0 & 1 & 0 \\ 0 & z_{hw}^{(2)} & 0 & 1 \\ \vdots & \vdots & \vdots & \vdots \end{bmatrix} \boldsymbol{a}^T \tag{7}$$

(8)

Let the above equation be expressed as $Y_{sim} = Z_{hw} a^T$ where $Y_{sim} \in \mathbb{R}^{2N \times 1}$, $Z_{hw} \in \mathbb{R}^{2N \times 4}$, and $a^T \in \mathbb{R}^{4 \times 1}$. Assuming we have sufficient number of pairs of corresponding points in simulation and real world, we can solve for $\boldsymbol{a}$ by least squares $a = (Z_{hw}^T Z_{hw})^{-1} Z_{hw}^T Y_{sim}$. The height $y_{sim}$ is chosen from a look-up table based on $y_{hw}$. Once we compute the transformation $A$, we store it for later to process arbitrary coordinates from real to sim, as shown below.

```python
def get_simulation_coordinates(xyz_hw: List[float], A: np.array) -> List:
    xz_hw = [xyz_hw[0], xyz_hw[2]]
    X_hw = get_homogenous_coordinates(xz_hw)
    X_sim_homo = np.matmul(A, X_hw)
    y_sim = process_height(xyz_hw[1])
    X_sim = [X_sim_homo[0]/X_sim_homo[2], y_sim, X_sim_homo[1]/X_sim_homo[2]]
    return X_sim
```

The objects used in simulation training are different from hardware objects, even though they belong to the same categories. For example, while both sim and real have a small plate, the sizes of these plates are different. We can estimate the size of the objects based on actual bounding box from the segmentation pipeline. However, it is significantly out-of-distribution from the training data, due to object mismatch. So, we map each detected object to the nearest matching object in simulation and use the simulation size as the input to the policy. This is non-ideal, as the placing might differ for sim versus real objects. In the future, we would like to train with rich variations of object bounding box size in simulation so that the policy can generalize to unseen object shapes in the real world.

## B   Simulation Setup

### B.1   Dataset

"Replica Synthetic Apartment 0 Kitchen" consists of a fully-interactive dishwasher with a door and two sliding racks, an adjacent counter with a sink, and a "stage" with walls, floors, and ceiling. We use selected objects from the ReplicaCAD [13] dataset – seven types of dishes (cups, glasses, trays, small bowls, big bowls, small plates, big plates). Fig. 11 shows a human demonstration recorded in simulation by pointing and clicking on the desired object to pick and place.

We initialize every scene with an empty dishwasher and random objects placed on the counter. Next, we generate dishwasher loading demonstrations, adhering to a given preference, using an expert-designed data generation script. Expert actions include opening/closing dishwashers/racks and picking/placing objects in feasible locations or the sink if there are no feasible locations left. Experts differ in their preferences and might choose different object arrangements in the dishwasher. A demonstration is a state-action sequence $\tau = \{(S_0, a_0), (S_1, a_1), \cdots, (S_{T-1}, a_{T-1}), (S_T)\}$. Here $S_i$ is the set of object and place instances and $a_i$ are the pick-place actions chosen by an expert at time $i$. At every time step, we record the state of the objects, the pick instance chosen by the expert, place instances for the corresponding category, and the place instance chosen by the expert. Expert actions are assumed to belong to the set of visible instances in $S_i$. However, different experts can exhibit different preferences over $a_i$. For example, one expert might choose to load bowls first in the top rack, while another expert might load bowls last in the bottom rack. In the training dataset, we assume labels for which preference each demonstration belongs, based on the expert used for collecting the demonstration. Given $K$ demonstrations per preference $m \in \mathcal{M}$, we have a dataset for preference $m$: $\mathcal{D}_m = \{\tau_1, \cdots, \tau_K\}$. The complete training dataset consists of demonstrations from all preferences: $\mathcal{D} = \bigcup_{m=1}^{M} \mathcal{D}_m$. During training, we learn a policy that can reproduce all demonstrations in our dataset. This is challenging, since the actions taken by different experts are different for the same input, and the policy needs to disambiguate the preference. At test time, we generalize the policy to both unseen scenes and unseen preferences.

## B.2 Expert Preferences

In Section 3, we define preference, that is, a set of 'properties' of the demonstration trajectory, like which rack is loaded first with what objects? Individual task preferences differ in the sequence of expert actions, but collectively, preferences share the underlying task semantics. For example, the user always opens the dishwasher rack before loading it for all preferences. By jointly learning overall preferences, our policy can benefit from cross-preference data to learn task structure, and sparse per-preference demonstration data to learn to follow the desired preference in the current situation.

There are combinatorially many preferences possible, depending on how many objects we use in the training set. Given 7 categories of dishes and two choices in which rack to load first, the hypothesis space of possible preferences is $2 \times 7!$. Our dataset consists of 12 preferences (7 train, 5 held-out tests) with 100 sessions per preference. In a session, $n \in \{3, ..., 10\}$ instances are loaded in each rack. The training data consists of sessions with 6 or 7 objects allowed per rack. The held-out test set contains 5 unseen preferences and settings for $\{3, 4, 5, 8, 9, 10\}$ objects per rack. Thus, there are $2 \times 7^n$ unique possible session demonstrations, i.e. different prompts and situations.

## B.3 Dynamically appearing objects

To add additional complexity to our simulation environment, we simulate a setting with dynamically appearing objects later in the episode. During each session, the scene is initialized with $p\%$ of maximum objects allowed. The policy/expert starts filling a dishwasher using these initialized objects. After all the initial objects are loaded and both racks are closed, new objects are initialized one per timestep to the policy. The goal is to simulate an environment where the policy does not have perfect knowledge of the scene and needs to reactively reason about new information. The policy reasons on both object configurations in the racks, and the new object type to decide whether to 'open a rack and place the utensil' or 'drop the object in the sink'.

## C Simulation Experiment Details: Baselines, Training and Metrics

GATs and Transformers are closely related network architectures. In theory, a GAT architecture over a fully-connected graph with an additional time-based feature would be equivalent to our Transformer-based model. We are certainly not the first to apply attention-based architectures to planning problems, but unlike [14, 11], we consider sequential planning over multiple timesteps,

which is critical to understanding human preferences [52]. Another difference from GNN for sequential planning is that we use cross-attention to condition the policy on the prompt, instead of adding preferences as an input feature. We use a slot attention architecture to learn a preference representation, and re-use instance encoders across prompt and situation. While these architectural choices are possible in GNNs, they are more intuitive and faster to implement in Transformers.

## C.1 Baselines

**Random** We sample the action, that is, pick or place instance from the set of visible instances in the current state/observation with equal probability. Similarly, the place pose is sampled randomly from all possible poses for the chosen category. The performance of a random policy reduces exponentially with the length of the action sequence (typically around 20-30 steps long), and number of visible objects in the scene (between 6 to 20). SPA is low if the policy either (1) fails to adhere to the physical constraints, or (2) violates the spatial preference. Even if we ignore the physical constraint violations and consider the task of selecting the preferred rack for each object, the probability of a SPA of 1.0 for a random policy would be around $(1/2)^{N_{objects}}$, since it has to choose either top or bottom rack per object.

**GNN-IL** We use GNN with attention-based message passing. The input consists of 12 dimensional attributes (1D-timestep, 3D-category bounding box extents, 7D-pose, 1D-is object bool) and 12 dimensional one-hot encoding for the preference $u$. This forms the input node features, with each node representing a visible object in the environment. All the nodes are fully connected with equal edge weight of 1. The preference vector $u$ is concatenated with each node $x_i{}_{i=1}^N$ of the state from the situation, resulting in 24 dimensional vector representation per node, which is in turn fed into the GNN decoder as used in [14]. Refer figure 12a for the overall architecture. The policy is trained using supervised learning to minimize the cross-entropy loss over the expert demonstration (choose the same object and place pose as selected by the expert of given preference).

**GNN-RL** We train the GNN decoder with Proximal Policy Optimization (PPO) [15]. Each node is represented similarly 12 instance attributes and ground-truth one-hot encoded preference (12-dimensional), like in GNN-IL. Reward function for the RL policy is defined in terms of preference. The policy gets a reward of +1 every time it predicts the instance to pick that has the category according to the preference order and whether it is placed on the preferred rack.

**GNN-VAE-IL** Note that GNN-IL gets privileged information about the preference encoding and not from the prompt demonstrations. So, we train a preference encoding based policy using VAE loss as per [11] and call this variant GNN-VAE-IL. The prompt is passed through the GNN encoder, global add pool and the feed-forward layers to predict mean $\mu$ and log variance $\sigma$ of preference latent $z$ as per [11]. This is used to sample a $z$ from preference embedding space to concatenate with the visible object instances in the situation. The concatenated preference to instance features is passed into the state-action policy modelled as GNN decoder in [14] to predict pick and place instances. GNN-VAE-IL is trained to minimize the VAE reconstruction loss over the actions taken by an expert of given preference in a situation.

## C.2 Architectural details

**TTP** We use a 2-layer 2-head Transformer network for encoder and decoder. The input dimension of instance embedding is 256 and the hidden layer dimension is 512. 100 slots and 3 iterations for Slot Attention. Instance attributes are encoded as follows: `Category` : $3 \rightarrow 64$, `Pose` : $7 \rightarrow 128$, `Timestep` : $1 \rightarrow 32$, `Is object marker?` : $1 \rightarrow 32$. For preference encoding, we use the slot attention layer at the head of Transformer encoder with 100 slots and 3 slot iterations.

**GNN Decoder** We use three layer GATConv, each followed by MLP for the GNN decoder with 128-dimensional hidden layer as per [14]. At the output, each node feature is reduced to a logit. We

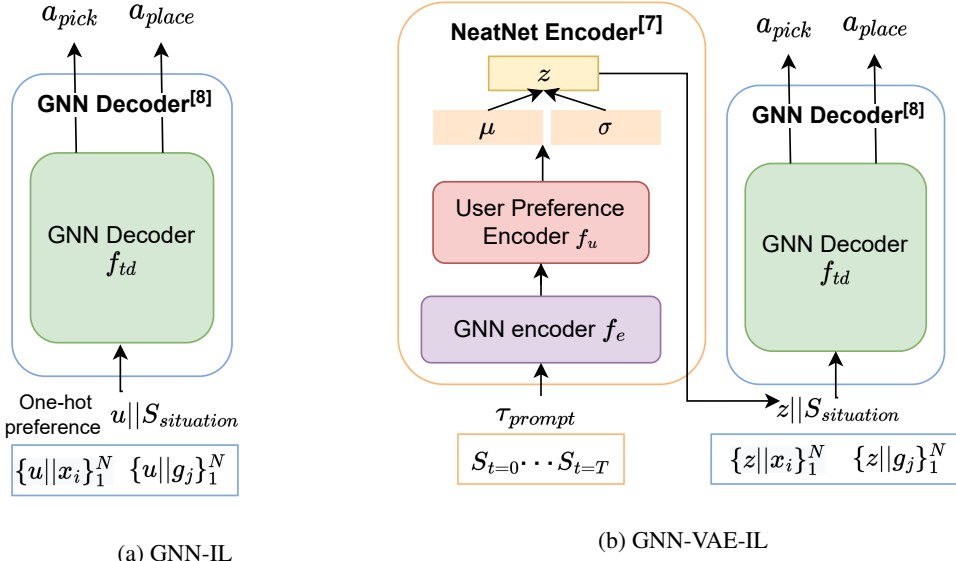

(a) GNN-IL

(b) GNN-VAE-IL

Figure 12: Model workflow for GNN-IL and GNN-VAE-IL inspired from [14, 11]. The one-hot encoding per preference is passed as input to GNN-IL or GNN-GT-IL (there is no unseen preference for GNN-GT-IL). GNN-VAE-IL infers the preference from prompt through reconstruction and KL loss on preference embedding space.

select the pick and place instance using the input 'is object marker', and consider the softmax from each respective set.

**GNN Encoder**   We use the GNN encoder as in [11] with 128 hidden dim GATConv layer followed by elu activation and 256 dimensional MLP. This is passed to a preference encoder, which is 2-layer MLP with leaky ReLU and that outputs 100-dimensional mean and log variance corresponding to the assumed Gaussian distribution of the inferred preference encoding. During training, the preference encoding is sampled from this distribution. While evaluation, we use the mean as the best representation of preference encoding.

## C.3   Training

We use a batch-size of 64 sequences. Within each batch, we use pad the inputs with 0 upto the max sequence length. Our optimizer of choice is SGD with momentum 0.9, weight decay 0.0001 and dampening 0.1. The initial learning rate is 0.01, with exponential decay of 0.9995 per 10 gradient updates. We used early stopping with patience 100. Fig.13 shows the accuracy achieved for picked category on uniformly sampled input states from the training data, unlike rollout where the next state depends on the previous action taken. We show runs grouped by batch sizes 64, 128 and 256.

## C.4   Metrics

In Section 3, we presented Spatial Preference Adherence (SPA) and Temporal Preference Adherence (TPA) metrics collected on a policy rollout. Here we discuss additional metrics about training progress and rollouts for GNN-VAE-IL.

**Category-token Accuracy**   indicates how well the policy can mimic the expert's action, given the current state. We monitor training progress by matching the predicted instance to the target chosen in the demonstration (Fig. 13). We see that TTP is able to predict the same category object to pick perfectly (accuracy close to 1.0). However, this is a simpler setting than sequential decision-making. During the rollout, any error in a state could create a setting that is out-of-distribution for the policy. Thus, category token accuracy sets

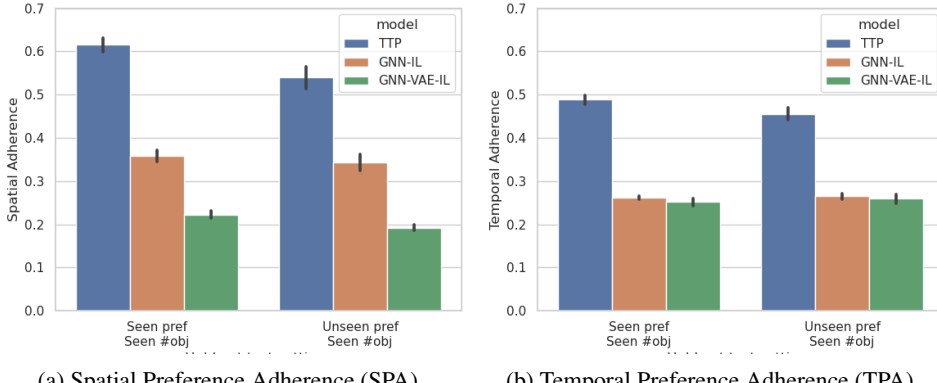

(a) Spatial Preference Adherence (SPA)  (b) Temporal Preference Adherence (TPA)

Figure 14: Comparisons of TTP, GNN-IL, and GNN-VAE-IL simulation across two metrics: SPA and TPA. TTP shows good performance at the task of dishwasher loading on seen and unseen preferences and outperforms the GNN-IL and GNN-VAE-IL baselines. Note that GNN-IL has ground-truth training data on unseen preferences.

an upper bound for rollout performance, that is, while having high category token accuracy is necessary, it is not sufficient for high packing efficiency and inverse edit distance.

**Policy Rollouts for Evaluation**  We evaluate trained policies on rollouts in the simulation, a more complex setting than the accuracy of prediction. Rollouts require repeated decisions in the environment, without any resets. A mistake made early on in a rollout session can be catastrophic, and result in poor performance, even if the prediction accuracy is high. For example, if a policy mistakenly fails to open a dishwasher rack, the rollout performance will be poor, despite good prediction accuracy.

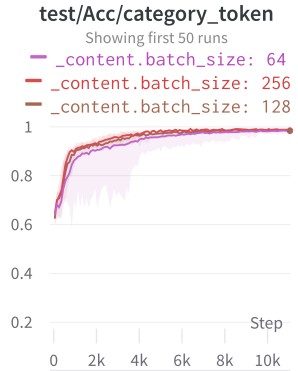

Figure 13: Category level accuracy grouped by batch size for prompt-situation training.

Note that the simulated dishwasher loading is a challenging task. The performance of the random agent is bad because we evaluate the SPA at the end of an episodic rollout. The performance of a random policy reduces exponentially with the length of the action sequence (typically around 20-30 steps long), and the number of visible objects in the scene (between 6 to 20). SPA is low if the policy either (1) fails to adhere to the physical constraints, or (2) violates the spatial preference. Even if we ignore the physical constraint violations and consider the task of selecting the preferred rack for each object, the probability of a SPA of 1.0 for a random policy would be around $(1/2)^{N_{objects}}$ since it has to choose either top or bottom rack per object.

We compare TTP, GNN-IL, and GNN-VAE-IL on seen and unseen preferences in Fig. 14. GNN-IL performs better than GNN-VAE-IL in terms of SPA but both are similar in terms of TPA. Intuitively, GNN-VAE-IL solves a harder learning problem of inferring preference from the trajectory, unlike GNN-IL which uses a privileged ground-truth label for the unseen preferences.

**Temporal efficiency**  Just like SPL [53] for navigation agents, we define the efficiency of temporal tasks in policy rollout, in order to study how efficient the agent was at achieving the task. For episode $i \in [1, ..N]$, let the agent take $p_i$ number of high-level interactions to execute the task, and the demonstration consists of $l_i$ interactions for the initial state. We scale the spatial preference adherence $SPA_i$ of the policy by the ratio of steps taken by the expert versus the learned policy. Temporal efficiency is defined between 0 to 1, and higher is better. This

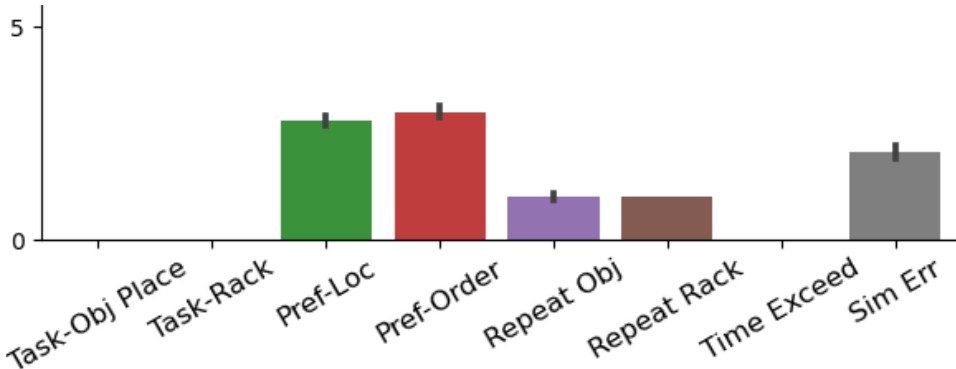

Figure 16: Failure analysis for TTP. X-axis indicates enlists different error types concerning physical constraints (**Task-Obj Place**: object place pose is infeasible, **Task-Rack**: not placing when in top when its open, not placing in bottom when top is closed and bottom is open, placing object when both racks are closed), preference constraints (**Pref-Loc**: location violation in terms of incorrect rack, **Pref-Order**: order violation of preference) and No-ops (**Repeat Obj**: picking an already placed object, **Reposition**: Adjust the placement of an object within the same rack, **Repeat Rack**: picking a dishwasher part repeatedly; leads to termination, **Time Exceed**: leads to termination). Y-axis is the error count per rollout. Most errors are due to failing in the location-aspect and order-aspect of the preference. The **repeat-*** pattern triggers the termination in most rollouts, indicating that failure scenario usually obeys physical constraints but does not make progress towards the task, thereby resulting in lower SPA and TPA.

value will be equal to or lower than the packing efficiency. This especially penalizes policies that present a 'looping' behavior, such as repeatedly opening/closing dishwasher racks, over policies that reach a low SPA in shorter episodes (for example, by placing most objects in the sink). Fig. 15 shows the temporal efficiency over our 4 main held-out test settings (as shown in Fig. 5).

# D    Failure Analysis

We present the failure analysis in terms of error counts over 63 rollouts. Figure 16 shows that TTP learns physical constraints of the problem perfectly, and never takes an action that violates them. Instead, the failure cases stem from either (1) not following user preferences, or (2) repeating certain actions which don't make progress towards the task,

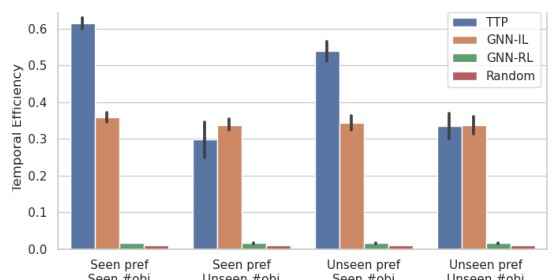

Figure 15: TE metric for held-out test settings.

like repeatedly picking and placing the same object. In contrast, our baselines like continuous pose prediction suffer from violating task constraints, like attempting to place an object in an already occupied location. Fig. 17 provides distribution per error type shown here, with leftmost bar near zero indicating that that error never occurs, and the right extreme denotes that that error occurs in high frequency in some rollouts. Both (1) and (2) stem out of the problem of sequential decision making. An open problem is to learn policies that can adapt from interaction and correct for such errors.

# E    Additional Ablation Experiments

Towards the end of the section 3.1, we presented ablation experiments over the number of demonstrations per preference used for training, and the number of unique preferences used. In this section,

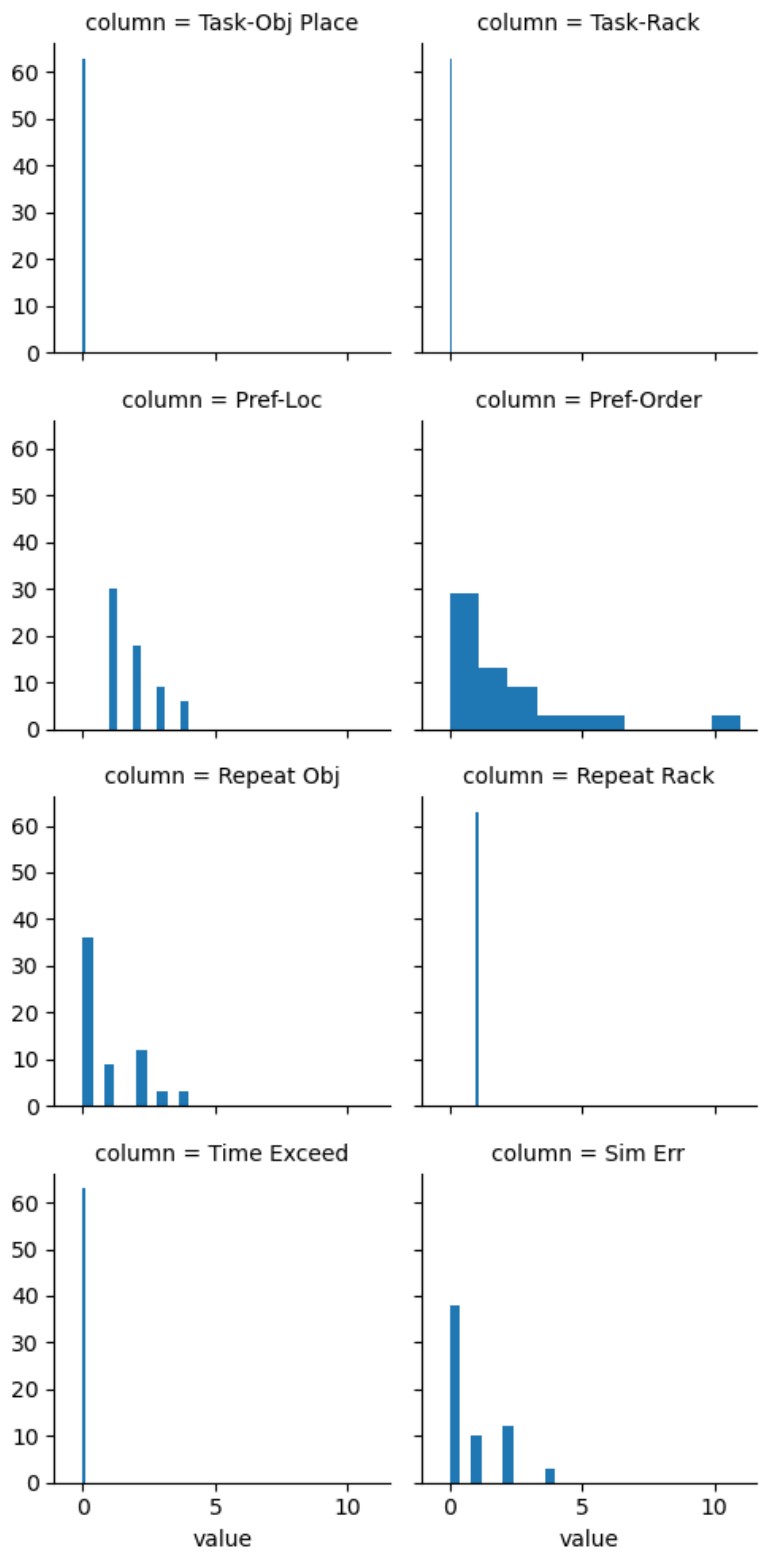

Figure 17: Histogram per error type to estimate the frequency (Fig 16 shows only mean and std). X-axis shows the value of the error count. Y-axis shows the number of times that value occurs in all of the rollouts. Most rollouts have 0 or very few preference based errors. Error count show peak near 0 and gradually reduces of towards higher numbers. This is an extended analysis of the error counts in figure 16 showing the failure profiles for the different error types.

we present additional ablation experiments over (E.1) the design of instance encodings in TTP, (E.2) the effect of increasing the temporal context of TTP on performance, the comparison between the choice of (E.3) discrete category tokens vs continuous box extent representation in TTP, and (E.4) continuous placement pose vs discrete alternative used in TTP.

## E.1 Design of Instance Encoding

**Why object attributes are important?**   The pose describes where an object is. For this task, it depends on whether the object is on the counter or in the dishwasher.

The category is important to follow the order-based preference and to group the placement instances. A big bowl may have very limited placement instances in a dishwasher rack compared to a plate or a small cup.

The timestamp is important for encoding the prompt – it allows the neural network to reason over prior instances without explicit tracking. Tracking an object across frames is difficult, especially with partially observed scenarios (such as an bowl in the top rack is not visible if the top rack is closed).

The $r$ marker is used to denote whether it is a pick or place instance. Marker acts as a flag and allows us to use the same architecture input to process a variable number of both pick and place instance embeddings.

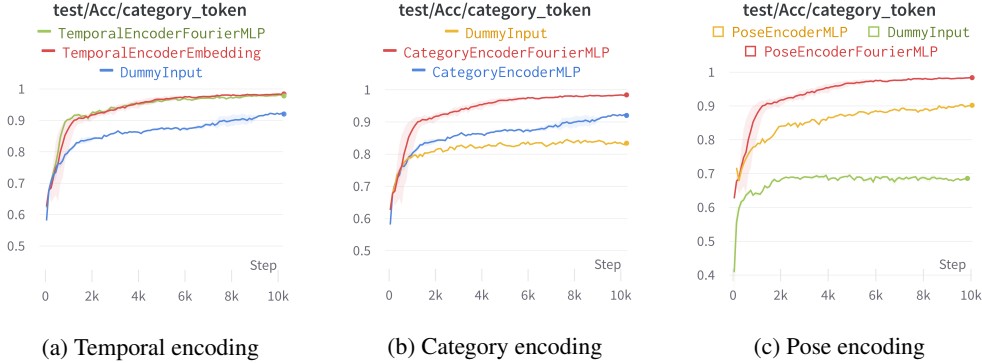

(a) Temporal encoding          (b) Category encoding          (c) Pose encoding

Figure 18: [Left-to-Right] Comparing different design choices of attribute encoders in terms of category token accuracy on held-out test prompt-situation session pairs.

**How much does temporal encoding design matter?**   Fig. 18a shows that learning an embedding per timestep or expanding it as Fourier transformed vector of sufficient size achieves high success. On the other hand, having no timestep input shows slightly lower performance. Timestep helps in encoding the order of the prompt states. The notion of timestep is also incorporated by autoregressive masking in both the encoder and the decoder.

**How much does category encoding design matter?**   In our work, we represent category as the extent of an object's bounding box. An alternative would be to denote the category as a discrete set of categorical labels. Intuitively, bounding box extents capture shape similarity between objects and their placement implicitly, which discrete category labels do not. Fig. 18b shows that Fourier transform of the bounding box achieves better performance than discrete labels, which exceeds the performance with no category input.

**How much does pose encoding design matter?**   We encode pose as a 7-dim vector that includes 3d position and 4d quaternion. Fig. 18c shows that the Fourier transform of the pose encoding performs better than feeding the 7 dim through MLP. Fourier transform of the pose performs better

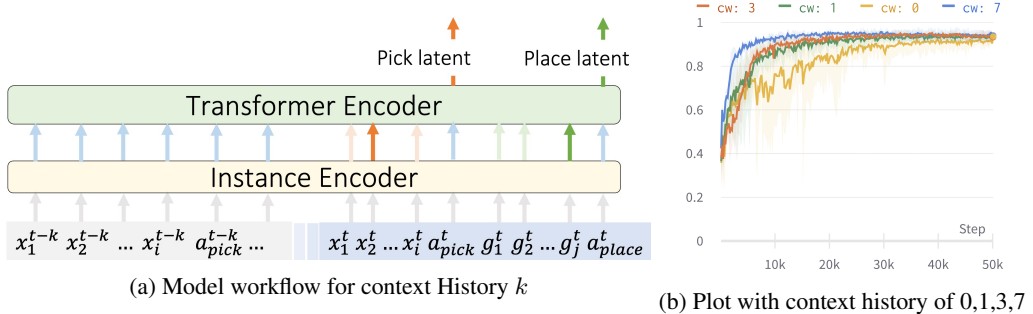

(a) Model workflow for context History $k$      (b) Plot with context history of 0,1,3,7

Figure 19: Incorporating Context History instead just current observation. **Left**: Modified TTP for processing with previous Context History $k$. **Right**: Plot showing category level accuracy for the held-out test sessions for single preference training with context windows. While larger context window size learns faster, the asymptotic performance for all context windows converges in our setting.

because such a vector encodes the fine and coarse nuances appropriately, which otherwise either require careful scaling or can be lost during SGD training.

## E.2 Markov assumption on the current state in partial visibility scenarios

Dynamic settings, as used in our simulation, can be partially observable. For example, when the rack is closed, the policy doesn't know whether it is full or not from just the current state. If a new object arrives, the policy needs to decide between opening the rack if there is space, or dropping the object in the sink if the rack is full. In such partially observed settings, the current state may or may not contain all the information needed to reason about the next action. However, given information from states in previous timesteps, the policy can decide what action to take (whether to open the rack or directly place the object in the sink). With this in mind, we train a single preference policy for picking only with a different context history. As shown in Fig. 19a, the context window of size $k$ processes the current state as well as $k$ predecessor states, that is, in total $k + 1$ states.

Let context history $k$ refer to the number of previous states included in the input. Then the input is a sequence of previous $k$ states' instances (including the current state), as shown in Fig. 19a.

Fig. 19b shows that TTP gets $> 90\%$ category-level prediction accuracy in validation for all context windows. While larger context windows result in faster learning at the beginning of the training, the asymptotic performance of all contexts is the same. This suggests that the dataset is largely visible and a single context window captures the required information. In the future, we would like to experiment with more complex settings like mobile robots, which might require a longer context.

## E.3 Discrete Category Tokens

Category embeddings should be more descriptive than just a bounding box and include some language labels like knife versus fork. In our current work, we assumed that similar-sized objects have similar occupancy footprints, and this was enough to reason about feasible placing locations in a rack. This was conducive to learning the physical constraints of objects in the scene but ignores other relationships between objects. Figure 20 shows TTP-DCat that indicates the performance with the discrete category is much lower and requires hyperparameter tuning for the embedding size. In future work, we would like to explore additional multi-modal object features, including language descriptions along with the geometry information.

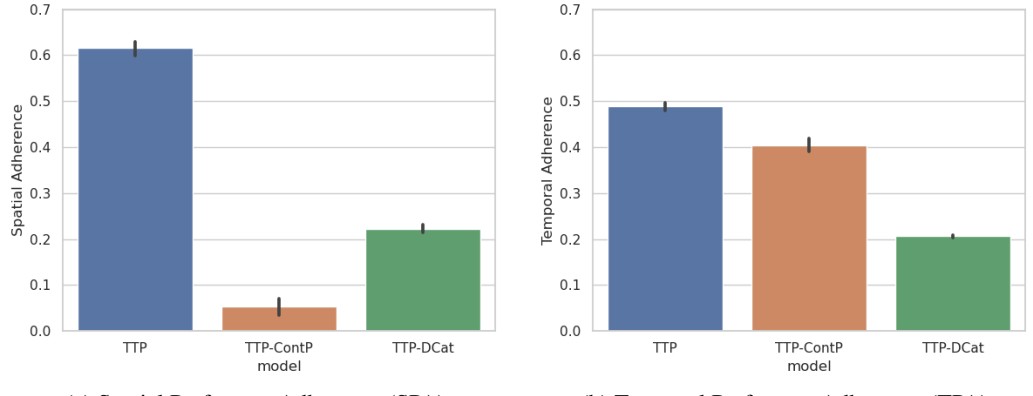

(a) Spatial Preference Adherence (SPA)  (b) Temporal Preference Adherence (TPA)

Figure 20: Comparisons of TTP, Continuous Place Pose Prediction (TTP-ContP), and Discrete Category Token Input (TTP-DCat), in simulation across two metrics: SPA and TPA (on Y-axis), on seen preferences and objects during training. Continuous Pose prediction achieves much lower SPA due to infeasible object place pose prediction and thereby violates the *location-aspect* of the preference. As the TPA is based on the pick order sequence, TPA is high indicating that the policy learns the *order-aspect* of the preference. Discrete category prediction achieves low SPA and TPA due to as few as 7 categories mapped to the embedding of size 64; similar to original TTP with continuous category representation in terms of the bounding box.

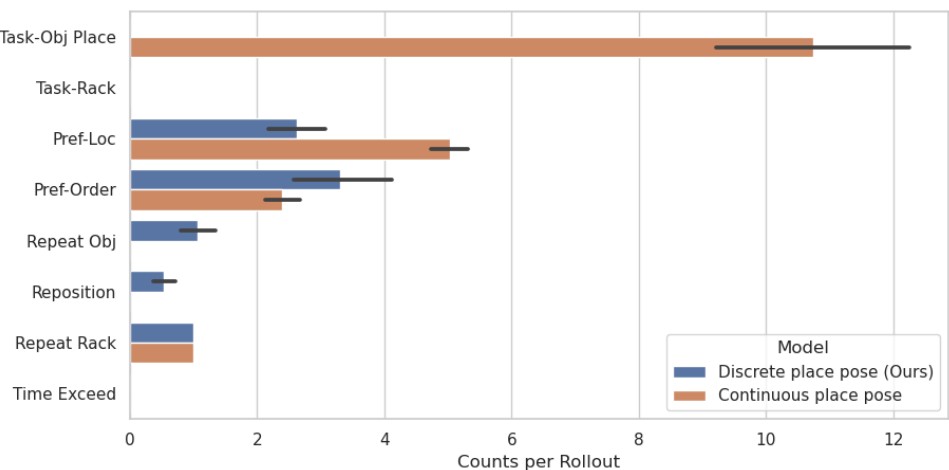

Figure 21: Failure case analysis for continuous place pose prediction. The X-axis indicates the error count per rollout. The Y-axis enlists different error types concerning physical constraints (Task-Obj Place: object place pose is infeasible, Task-Rack: not placing when in the top when its open, not placing in the bottom when the top is closed and the bottom is open, placing object when both racks are closed), preference constraints (Pref-Loc, Pref-Obj) and No-ops (Repeat Obj: picking an already placed object, Reposition: Adjust the placement of an object within the same rack, Repeat Rack: picking a dishwasher part repeatedly; leads to termination, Time Exceed: leads to termination).

### E.4 Continuous Placement Pose

In TTP, we consider two solution spaces: decision over 'what to pick', and 'where to place'. The pick solution space is naturally discretized by the objects of interest visible in the environment (dishwasher, racks, dishes, etc.), and does not limit real-world transfer.

Note that while we consider discrete objects, their features like location are continuous. An object segmentation pipeline, as used in our real-world experiments, can provide a discrete set of objects to interact with, and their continuous features. On the other hand, our choice of discretized 'where to place' solution space can be seen as a hurdle to real-world transfer. However, in our experiments, dense discrete poses generalized to unseen scenarios better than continuous prediction. Figure 20 shows that discrete placement pose instances achieve higher SPA and TPA than continuous place pose prediction. The TPA looks comparable to the original TTP but the SPA is much lower. This is because the TTP-ContP policy learns to pick objects in the correct order like TTP but suffers in placement pose prediction. Figure 21 shows the count of error types over 9 rollouts for each of the 7 preferences (=total 63 rollouts). Here lower error count is better. Error for continuous place pose policy occurs mostly in the prediction of infeasible place pose for the object.

The geometry of a dishwasher is well-suited for discrete positions, as small prediction errors can make a target place position infeasible. For example, a plate can only fit in a particular pose, and small orientation errors make placing infeasible. In the real world, one could imagine creating a dense place pose dictionary based on the model of the dishwasher in the user's home. However, we agree that for other receptacles, like a shelf, this discretization is not necessary, and a continuous prediction approach like our baseline might be better suited.

Finally, we note that while we use discrete place poses, they are densely sampled, and not mutually exclusive. Hence, several discrete poses overlap, and if one pose is occupied by an object, there might be several others that are also occupied by the same object. TTP has to reason about the size of an object and distances between place poses to decide which pose is empty to place the next object, while maximizing packing efficiency. Hence, our policy is able to reason about 3D space occupancy, even if working with discretized poses.

## F Limitations and Future scope

In Section 5, we briefly discussed the limitations and risks. Here we enlist more details and highlight future directions.

**Pick grasping depends on accurate segmentation and edge detection**   Grasping policy depends on the quality of segmentation and edge detection of the selected object. Due to noise in calibration, shadows, and reflections, there are errors in detecting the correct edge to successfully grasp the object. For example, it is hard to grasp a plate in a real setting. The plate is very close to the ground and the depth cameras cannot detect a clean edge for grasping. As the depth edges are not often not accurate due to noisy point clouds, the collision prediction with the candidate grasps and the object is not very robust and conservatively results in no feasible grasps. Therefore, in our work, we place the plate on an elevated stand for easy grasping. Grasping success also depends on the size and kind of gripper used.

**Placement in real setting**   For placement, the orientation of the final pose is often different from the initial pose and may require re-grasping. The placement pose at the final settlement is different from the robot's end-effector pose while releasing the object from its grasp. Similar to picking, placement accuracy will largely depend on the appropriate size and shape of the gripper used. Due to these reasons, placement in the real world is an open challenging problem and we hope to address this in future work.

**Hardware pipeline issues due to calibration**   The resulting point cloud has noisy depth edges due to two reasons. First, incorrect depth estimation due to camera hardware, lighting conditions,

shadows, and reflections. Second, any small movements among cameras affect calibration. If we have a noisy point cloud, it is more likely to have errors in subsequent segmentation and edge detection for grasp policy. Having sufficient coverage of the workspace with cameras is important to mitigate issues due to occlusions and incomplete point clouds.

**Undesired motion due to IK and grasping**  We use the Pinnochio library [54] for planning robot joints given the end-effector as this is fast and easy to use. But it assumes collision-free plans, which is difficult with relaxed IK limits to enable robot arm is reachable to the base of the drawers. Also, the robot faces difficulty in grasping certain objects that are closer to the base and occluded in the camera images.

We use GraspNet [21] for obtaining analytical candidate grasps on the segmented point cloud for a dish. The best candidate grasp may not be always semantically appropriate, for example, the best candidate grasps on a cup or wine glass are closer to the pinched at the rim (due to clearer edge detection) but many humans prefer to hold the curved cylinder stem. Incorporating preferences in low-level grasping is an open direction for future work.

**Incomplete information in prompt**  The prompt session may not contain all the information to execute the situation. For example, in a prompt session, there might be no large plates seen, which is incomplete/ambiguous information for the policy. This can be mitigated by ensuring complete information in the prompt demo or having multiple prompts in slightly different initialization.

