# OpenReview forum: "Transformers Are Adaptable Task Planners"
_robot-learning.org/CoRL/2022/Conference — CoRL 2022 Poster_

### Official Review · Reviewer_qr8n · 2022-07-03

**Originality:** Good
**Technical Quality:** Excellent
**Clarity Of Presentation:** Excellent
**Impact:** 3

**Recommendation:**

Weak Accept: I recommend accepting the paper, but will not argue for my recommendation if the majority of other reviewers have a different opinion.

**Summary:**

This work proposes a transformer-based approach for learning preference-conditioned policies in object-centric manipulation environments (specifically, loading a dishwasher). Preferences are expressed through single demonstrations. Learning is done via behavioral cloning on a given set of preference-labeled demonstrations. Experiments in simulation and on a real robot compare the proposed approach with graph neural network baselines and consider generalization to unseen numbers of objects and unseen preferences.

**Issues:**

   * I think that use of the word “planning” in the title and name of the approach is misleading. Planning usually has the connotation of imagining possible futures with a transition model and reasoning toward a goal. The approach in this work uses a reactive policy and does not use transition models or goals.
   * The word “category” is also potentially misleading, because if I understand correctly, the category is just the bounding box of the object. A better term might be “shape” or “size” or “geometry”. I like the additional analysis and discussion in Figure 8b, though.
   * The definition of the encoded state on L85 suggests that the instances are ordered into a list before passing through the network. Is the model invariant to this order? It should be, right? If so, I would just suggest changing the list to a set in the notation.


### Minor
   * L16: “task constraints through symbolic task description” -> descriptions
   * The x_i start at 0 on L61 but started at 1 on L56
   * I found the term “situation” confusing in Section 2.1. For example, in figure 3, “Situation” is input to the instance encoder on the bottom. But isn’t that just the current state? In other places, it seems like situation is meant to refer to both the state and the preference. Also the “situation decoder” and “policy” are maybe synonymous?
   * L112: “The left half is prompt encoder” → the prompt encoder
L113: “acting on given state” → “the given state”
   * L124: “as well as, placement” → “as well as placement”
   * Figure 4 caption missing a period
   * L172: “unseen number of object” → “numbers”
   * Figure 5 caption: “if worse” → “is worse”
   * It would be good to make the y axis limits identical in Figure 6
   * I didn’t understand what this means: “The policy was successful once with 100% success rate, and once with 75%”
   * There are broken figure and section references in the appendix


**Quality Of The Limitations Section:**

Limitations are addressed clearly

**Reviewer Expertise:**

3: The reviewer is fairly confident that the evaluation is correct

**Robotics Focus:**

Sufficient demonstration on hardware

**Strengths And Weaknesses:**

### Strengths

* I really like the “dynamically appearing objects” setting in the experiment and appreciate the in-depth analysis in the appendix.
* I really appreciate the detailed experimental analysis, including the results in Figure 6, Table 1, and in the appendix. The experimental design overall is great.
* The writing overall is very clear and easy to understand.
* The related work seems thorough.
* The dishwasher application is nice and clear, and seems like a challenging application.


### Weaknesses

* My biggest concern is the representation of preferences that is given to the GNN baselines. L198 states that the representation is categorical and argues that this is privileged information. For the in-distribution experiments, this is maybe okay, but there is still considerable information loss in using categories versus full demonstrations. But for the out-of-distribution experiments, do I understand correctly that the GNN will simply receive a different categorical value than it saw during training? There would be no hope for generalization if this is the case, and this would be an unfair comparison to TTP.
* It is a little disappointing that generalization to out-of-distribution object numbers is poor, but I appreciate the analysis in Figure 6a on this point.
* The work is a little more incremental than novel, but the all-around quality and thoroughness still makes for a solid contribution.
* There is some redundancy in Section 2 that could be removed. For example, in the State-Action Representation subsection, there is discussion of the policy. But then the policy is discussed again throughout the rest of the section. Section 2.2 is also somewhat redundant with the end of Section 2.1. Overall, more strict adherence to the topics of the subsections would help clarity.
* If I understand correctly, the demos used in the main results are generated as follows: K demos are generated for loading the dishwasher in order A, B, C…; then K more demos are generated for loading the dishwasher in order C, A, B, …; etc. This procedure seems pretty far removed from a real LfD setup. In reality, I would expect that humans loading dishwashers may have some preferences, but are also largely ambivalent about order. Furthermore, any particular human would probably be inconsistent if you asked them to load the dishwasher multiple times. It would be nice to see results for demos collected from real users, or at least demos collected in a noisier and less structured way.

**Summary Of Recommendation:**

This work is limited in novelty, but otherwise very well done. I think the in-depth experimental analysis and the details of the policy architecture will comprise nice additions to the literature.

---

> ### Author Response · Authors · 2022-08-26
> **Author Rebuttal (1/2)**
>
> We are glad that the reviewer found our “experimental design overall great”, “challenging application”, and “in-depth analysis in the appendix”. Our “writing overall is very clear and easy to understand” and “related work seems thorough”. We address some concerns below:
>
> > My biggest concern is the representation of preferences that is given to the GNN baselines.
>
> Thank you for this comment. First, we want to clarify that there are no out-of-distribution preferences for GNN with categorical input. The GNN-IL baseline in our paper was trained on all 12 preferences, including the unseen ones. Hence there is no need for the GNN policy to generalize to preferences, and “the GNN will simply receive a different categorical value than it saw during training? “ is incorrect, because the GNN sees a dataset for all preference categories during training. On the other hand, TTP only sees a subset of preferences (7) during training, and  generalizes to 5 unseen preferences at test time. Hence this is privileged information for the GNN over TTP. Despite this, on unseen preferences, TTP outperforms GNN on seen #objects (SPA 0.62 with TTP vs 0.34 with GNN), and performs comparably on unseen #objects (SPA 0.34).
>
> Second, we add an additional experiment to the Appendix C Figure 14 where we train a preference representation together with the GNN policy using the approach from [8]. The preference representation is trained to reconstruct the action taken by the expert in the demonstration, and has exactly the same information as TTP. However, training of this model is slow, and we are able to present only preliminary results on a single seed. On this seed, we see that both TTP, and GNN-IL with ground-truth categorical information (GNN-IL-GT) outperform the new baseline (GNN-IL-VAE). Preliminary results are: SPA of 0.54 for TTP, vs SPA of 0.34 for GNN-IL-GT, vs SPA os 0.19 for GNN-IL-VAE on unseen preferences and seen #objects.
>
> We hope that the additional explanation and the added experiment addresses the reviewer’s concern about the GNN baseline. We have also updated our paper to better explain the training of GNN-IL-GT, and GNN-IL-VAE models and will update GNN-IL-VAE results once more runs are completed.
>
> > There is some redundancy in Section 2 that could be removed
>
> Thank you for pointing this out. We have updated the paper and will improve it to make the flow smoother.
>
> > I would expect that humans loading dishwashers may have some preferences, but are also largely ambivalent about order. Furthermore, any particular human would probably be inconsistent if you asked them to load the dishwasher multiple times.
>
> Thank you for this insightful comment. We agree that users might be ambivalent about some aspects of the task, or even inconsistent with demonstrations. Right now, our approach is not able to deal with such situations where the demonstrations are inconsistent, or incomplete in terms of preference. We think that the solution for this is moving towards a weakly-supervised or unsupervised learning approach with a larger dataset. The policy could be pre-trained using a contrastive loss [a], or to maximize a diversity objective like [b], and fine-tuned on a small scale (single?) demonstration. We leave this to future work.
>
> ---
> ### References
>
> *[a] Chen, Ting, et al. "Big self-supervised models are strong semi-supervised learners." Advances in neural information processing systems 33 (2020): 22243-22255.*
>
> *[b] Eysenbach, Benjamin, et al. "Diversity is all you need: Learning skills without a reward function." arXiv preprint arXiv:1802.06070 (2018).*

---

> ### Author Response · Authors · 2022-08-26
> **Author Rebuttal (2/2) + Plots attached**
>
>
>
>
> > The word “category” is also potentially misleading
>
> We agree that currently the category embedding only uses a bounding box, and hence is closer to “size”. However, we hope that in the future it can use more descriptive features about the object, for example, language descriptions like spoon, knife, etc, or visual features like a segmented image. This can help the policy reason about similar items like forks and spoons, but also about object properties like plastic items should go on the top rack.
> We hope this convinces the reviewer of our motivation for using “category” instead of “size” or “geometry”, but if this is still an issue, we will change the name in the final paper.
>
> Additionally, we added empirical results on discrete category representation in the current setting with 7 categories. These may improve with hyper-parameter tuning for embedding size, but it is not performing well when compared with the same model size as the original TTP. Refer Figure 20 in Appendix E.3.
>
> > instances are ordered into a list..changing the list to a set in the notation
>
> You are correct, and thank you for pointing out this error in our notation. We have fixed this in Section 2 and corrected the other typos you mentioned in the paper.
>
> We hope that the additional baseline experiment with GNN-VAE-IL in Appendix D and our updated paper have addressed most of the reviewer’s concerns. Please let us know if there are any other concerns, and we will be happy to update our paper.
>
> > use of the word “planning” in the title and name of the approach is misleading
>
> We understand the reviewer’s concern, but also disagree with this comment. Neural-network policies have been shown to present characteristics of model-based planners before [c].  Such policies are able to reason about the future (possibly by building implicit models of the world), generalize to unseen settings, make long-horizon decisions like moving away from the goal in non-minimum-phase systems, to achieve a task. For example, while the final goal state of the dishwasher is closed with objects in it, TTP decides to open the dishwasher and racks to start placing objects. In this way, it exhibits the same behavior as a planner designed for the task of loading the dishwasher with visible dishes. We use this behavior to argue that TTP is indeed a planner, and it is reasoning about the world, and its future states implicitly in making decisions.
>
> ---
> ### References
>
> *[c] Guez, Arthur, et al. "An investigation of model-free planning." International Conference on Machine Learning. PMLR, 2019.*

---

### Official Review · Reviewer_3v7A · 2022-07-27

**Originality:** Very Good
**Technical Quality:** Very Good
**Clarity Of Presentation:** Excellent
**Impact:** 4

**Recommendation:**

Strong Accept: I recommend accepting the paper and will argue for my recommendation even if other reviewers hold a different opinion.

**Summary:**

This paper uses a household task "placing kitchen objects in the dishwasher" to test a newly proposed preference-guided task planner model using a transformer. The authors applied plenty of techniques from the natural language processes in the current work to encode the environment input and embed human preferences, which is very interesting and brings good inspiration to the robotics community. The work is both theoretical and technical and the experiment result has been well discussed.

**Issues:**

- I would ask the authors to improve fig.3 to add more technical details in it, see the weakness section.
- I would ask the authors to add some reasons for why choosing GNN as a baseline and an illustration of how the baseline has been used in the current setups.
- Figure 5, caption, typo: "if worse"
- There are a few small errors in the appendix, like missing hyperlinks.
- Figure 5b, why does the GNN-IL perform better in unseen preferences and unseen objects?
- My understanding of human demonstrations is a long sequence of which object to take and where to place it, which is the high-level action choices, and it is also the main focus of the current paper. But, how can we get the low-level control commands, and deal with the issues such as collision avoidance?
- In line 225, the performance on 3-4 objects is poorer and the authors claimed it is because it is unseen. But my question is that if e.g. 5 objects have already been placed in the dishwasher from the setup with 8 objects initialized at the beginning, it reached a similar situation having 3 objects as the initial state. So what are the differences between the two cases?

**Quality Of The Limitations Section:**

Limitations are addressed clearly

**Reviewer Expertise:**

4: The reviewer is confident but not absolutely certain that the evaluation is correct

**Robotics Focus:**

Sufficient demonstration on hardware

**Strengths And Weaknesses:**

Pros:
- The motivation of long-term motion sequence planning is well explained. The reason to use a transformer as a planner is clear and convincing.
- The conception, model, and math symbols are structured and explained in a very clear way. It is very straightforward to capture the workflow and ideas from the paper, even for people without much experience in transformer and sequence planning, like me.
- The paper is well written. I enjoy reading it without finding any explicit typos or mistakes.
- The authors have offered a comprehensive prior work investigation, including sequential manipulation, transformers as sequence modeling, and preferences and prompt training, which benefits the readers to work in a similar field.
- The authors used the task "putting kitchen bowls and plates in a dishwasher" to prove their model's advantages. In my opinion, such a setup is a good choice and is complex enough to prove the advantage of the model.
- This paper has offered a proper ablation study to prove all the attributes (pose, time, category) are necessary to achieve a good result.
- Interestingly, the authors used some knowledge from the NLP field in the current robotic task to support the good performance of the transformer model, which breaks the barrier between the two fields and offers some inspiration to the researcher in the robotics field who do not have much experience in the NLP.
- Due to the page number limitation, some of the details regarding the experiment settings, type of preferences, the reason for using drawer in reality, camera perception, etc. are missing in the main manuscript but offered in the appendix, which has answered some of my questions. The demo video is also nicely made and brings a lot of useful information.

Cons:
- It will be great if the authors can extend figure 3 and illustrate the content from lines 122 to 134 in it as well. This will help understand the mechanism details.
- The author presented in the prior work section, that there is a lot of related work in this field. But unfortunately, I do not see a clear reason to use GNN-based preference learning as a baseline.
- In contrast to the proposed model, however, I don't know how the GNN baseline works, especially because the baseline is a combination of two different papers on GNN. I would suggest the authors add some illustrations in the appendix to show the workflow on page 6 of the baseline as well, as to improve the confidence of the readers.


Neutral:
- A nitpicky question regarding the preferences, but I expect an answer from the authors. Sometimes we as users do not know our preferences. Badly, we are also lazy to perform any demonstration to show our preferences. So is it possible to have some pre-programmed preferences mode, e.g., "place bowls first on the higher rack, then put plates on the lower rack"? Because the preference embedding is an intermediate variable in the current model, it is hard to translate it to "mode 1, mode2, ..." and ask the user to choose.



**Summary Of Recommendation:**

I recommend this paper as l enjoy reading it, although I am neither a researcher in human preference learning nor attention mechanism. This paper focuses on a very practical problem that needs a high-level sequence learning conditioning human preferences. The model is well explained and the experiment is clearly presented and discussed. I can capture most of the ideas of this paper as well as recognize the main contribution of this paper. Although there are a few unclear things that I have pointed out in the weakness section, I believe the authors are capable to improve the paper in the revision session to strengthen my confidence even further. However, if the other reviewers with more knowledge in Attention or Preference learning pointed out some critical issues of the current paper, I will also consider their arguments.

---

> ### Author Response · Authors · 2022-08-26
> **Author Rebuttal (1/2)**
>
> We are encouraged by the positive feedback from the reviewer, and that they “enjoy reading it”. We are glad the reviewer found our motivation “well explained”, and “the reason to use a transformer as a planner is clear and convincing”. Our model and workflow was “explained in a very clear way” and “straightforward to capture”, “prior work investigation comprehensive”, dishwasher-loading “a good choice and complex enough”, and “breaks the barrier between the two fields– robotics and NLP”. We address the concerns raised by the reviewer below:
>
> > extend figure 3 and illustrate the content from lines 122 to 134
>
> We have updated Figure 3 to include the inputs and notation to each part of the model. We hope that Section 2 on prompt-situation architecture is more readable and relatable with the notation in the figure.
>
> > I do not see a clear reason to use GNN-based preference learning as a baseline.
>
> Thank you for this comment, we have updated our related works section to better highlight why other prior work is not a suitable baseline. In summary, most prior works do not deal with long-horizon planning tasks with task constraints and preferences, except TransporterNets [31] and SORNet [32], and our GNN baselines [7, 8]. [31] does not consider generalization in sequential reasoning problems, and primarily deals with Pick&Place generalized to new object locations. SORNet assumes access to a task planner, which reasons about sequential reasoning instead of a learned policy. Such a task planner can be tedious to design for complex tasks, and difficult to modify per preference. Instead, [7] presents a learned sequential reasoning approach using GNNs, and [8] shows generalization of GNNs to preferences. Hence, we combine [7] and [8] to create a sequential reasoning baseline that can generalize to preferences.
> We hope that the updated related works with this description will make our choice of the GNN-based baseline more clear. Please let us know if you have more comments, and we would be happy to incorporate them.
>
> ---
> ### References
> *[7] Y. Lin, A. S. Wang, E. Undersander, and A. Rai. Efficient and interpretable robot manipulation with graph neural networks. IEEE Robotics and Automation Letters, 2022.*
>
> *[8] ​​I. Kapelyukh and E. Johns. My house, my rules: Learning tidying preferences with graph neural networks. In Conference on Robot Learning, pages 740–749. PMLR, 2022.*
>
> *[31] A. Zeng, P. Florence, J. Tompson, S. Welker, J. Chien, M. Attarian, T. Armstrong, I. Krasin, D. Duong, V. Sindhwani, and J. Lee. Transporter networks: Rearranging the visual world for robotic manipulation. Conference on Robot Learning (CoRL), 2020.*
>
> *[32] W. Yuan, C. Paxton, K. Desingh, and D. Fox. SORNet: Spatial object-centric representations for sequential manipulation. In 5th Annual Conference on Robot Learning, 2021. URL https://openreview.net/forum?id=mOLu2rODIJF.*

---

> ### Author Response · Authors · 2022-08-26
> **Author Rebuttal (2/2) + Plots attached**
>
> > I would suggest the authors add some illustrations in the appendix to show the workflow on page 6 of the baseline as well
> >
>
> Thank you for this suggestion! We have added the figure of the baselines to our appendix C Figure 12 where we describe all baseline approaches, including the GNN baseline in more detail. We hope that this additional description makes the reviewer more confident in our experiments.
>
> > is it possible to have some pre-programmed preferences mode, e.g., "place bowls first on the higher rack, then put plates on the lower rack"?... it is hard to translate preference to "mode 1, mode2, ..." and ask the user to choose.
> >
>
> Indeed, currently our model depends on a prompt demonstration. So to encode different default preferences, we would need to pre-build a look-up table that maps modes to a prompt demonstration. The user can then choose from a range of “pre-sets” to choose one of the modes, which can be used for conditioning the policy.
> Ideally, we would like to condition our policy on natural language based preferences, so that it is easier for users to interpret a preference, as well as specify a variant and would explore further in future work.
>
> > Why does the GNN-IL perform better in unseen preferences and unseen objects?
> >
>
> The GNN-IL baseline is trained on all preferences, including “unseen preferences”. This is privileged information over our approach TTP, which is being tested on unseen preferences. Despite this privileged information, GNN-IL performs very similarly to TTP (SPA 0.347 with GNN-IL vs 0.338 with TTP). This was originally explained in Section 3.2 in our paper, and we additionally added it in the description section for Figure 5.
>
> > How can we get the low-level control commands, and deal with the issues such as collision avoidance?
> >
>
> Currently, we use very simple motion-planning to achieve low-level robot motions. Specifically, we plan a very sparse set of waypoints (hand-designed for now, but can be replaced by something like RRT) that can let the robot move in the environment without collision. This aspect is not learned from demonstrations, or transferred from sim-to-real.
>
> > if e.g. 5 objects have already been placed in the dishwasher from the setup with 8 objects initialized at the beginning, it reached a similar situation having 3 objects as the initial state. So what are the differences between the two cases?
> >
>
> While the number of objects on the counter are the same in the two cases, there are more objects already placed in the dishwasher in the session with 8 objects, unlike a session with 3 objects. Since the visible objects already placed in the dishwasher are also added to the policy input, the setting with only 1 or 2 objects on the counter, and the dishwasher empty is out of distribution for the policy. Ideally, we would like the policy to update from interaction and learn to deal with such related, but slightly different scenarios, through experience.

---

### Official Review · Reviewer_jkUB · 2022-07-31

**Originality:** Good
**Technical Quality:** Good
**Clarity Of Presentation:** Good
**Impact:** 2

**Recommendation:**

Weak Accept: I recommend accepting the paper, but will not argue for my recommendation if the majority of other reviewers have a different opinion.

**Summary:**

This paper presents a transformer-based approach to incorporating human preferences for high-level task planning. The authors focus on simulated and real-world dishwasher loading tasks, in which humans have preferences about where items should be loaded and the order in which loading occurs. Preferences are communicated to the robot through a single demonstration. The proposed approach demonstrates an ability to understand those preferences from that demonstration. The proposed approach improves on an imitation-learned baseline (using a GNN) trained by the authors in situations where the number of objects matches that seen during training and is competitive otherwise.

**Issues:**

The authors should focus on clarifying both the technical details of their approach as well as better motivating the problem setting and experiments presented in the paper. In addition, experiments or results that show the impact of expecting the agent to predict actions to open/close the dishwasher and doors will greatly help to understand the efficacy of the proposed approach on learning preferences without the need to also learn manipulation constraints.

**Quality Of The Limitations Section:**

Limitations are addressed clearly

**Reviewer Expertise:**

2: The reviewer is willing to defend the evaluation, but it is quite likely that the reviewer did not understand central parts of the paper

**Robotics Focus:**

Sufficient demonstration on hardware

**Strengths And Weaknesses:**

The paper has weakness stemming from two main areas: the motivation behind the chosen experimental scenario and clarity regarding the technical approach. [I will also admit that the clarity of the paper could also use some improvement; it is possible that I have misunderstood some key aspects of the paper, and have incorporated clarification questions in my discussion below so that the authors can help me better understand if I am incorrect.]

First, I have concerns about the central metric being used for evaluation (or the clarity with which it is presented): the Packing Efficiency. The definition of the Packing Efficiency must be clearer, as it is difficult right now to understand how it is computed yet is key for understanding the performance of the proposed approach. Do a_hat and b_hat incorporate the preferences, or is the equation on lines 185--186 computed first using the items placed in the dishwasher and *then* the result is set to zero if the preferences are not met? Does this mean if *any* preferences are violated, the PE is zero. If so, this would help to explain why the performance of the random agent is so low, since it would require getting quite lucky to achieve a result in which all the objects are placed correctly. If not, it is not clear why the performance of the random agent would be so poor, since if the dishes were randomly placed, should not the performance of the system be roughly 50% accurate? Either way, some clarity on the random baseline and the precise meaning of the PE metric would greatly help the paper.

The experimental setup is not particularly well justified in the text. Critically, the agent's tasks include opening and closing the dishwasher and sliding racks in and out. The authors mention in a few places in the text that planning failures may occur when an agent attempts an action to put an item in the dishwasher yet the dishwasher is closed. There are two issues with this.

First, the rate of failures associated with constraints borne of these manipulation constraints is not reported in the text, making it difficult to understand the impact they have on the performance. The authors should either better justify their decision to incorporate these actions/constraints or provide statistics along with their existing results that identify the rate of failures associated with physical constraints. The purpose of the proposed approach is to evaluate the ability of the agent to identify and incorporate preferences, yet it seems that many of the demonstrations are limited instead because the system is additionally asked to predict physical constraints.

Second, expecting the agent to explicitly enumerate those actions feels like an artificial limitation; the agent is already running a non-learned deterministic planner to plot the trajectories to move the dishes into the dishwasher, so it is unclear why that planner would also not be able to ensure that those constraints are first met. On line 173, the authors point out that these manipulation constraints are why the problem setting is more challenging than a simple prediction task. However, if the motion planner takes care of those constraints simply predicting where objects should be placed in the dishwasher may be able to achieve high performance on the PE metric (though not the Inverse Edit Distance metric). In the text, the authors must better justify the nature of this problem setting or clarify why the proposed methodology is the way to best evaluate this system, since it is important to motivate the need for their approach to be used.

Finally, it is not clear why the order in which objects are loaded into the dishwasher is a particularly salient concern, and so the authors should clarify why dish loading preferences need to include order of operations in addition to placement region. This is the smaller of my high-level concerns, since I am willing to accept that there are household scenarios in which the sequence of operations matters. I am not convinced that loading dishes into the dishwasher is one of those tasks. However, the results showing that it is *possible* using the proposed approach are still valuable, and I think do add to the contributions of the paper. Rather than performing new experiments, which I do not think necessary for this purpose, the authors should discuss a situation in which order of operations is a preference that robots should consider when executing household tasks (even if that scenario does not appear in the paper).

Smaller questions and comments:
- Why is a bounding box used as an input for the category label rather than a one-hot encoding of the object? This seems prone to error and feels like an artificial limitation. While additional experiments are unlikely to occur, the authors should better justify this decision.
- The equation for PE seems to have an error; if the robot's performance is perfect, each of the terms inside the argument should be 1 (totaling to 2) and then summed over the individual demonstrations.


**Summary Of Recommendation:**

The paper was somewhat hard to follow in places, as some details (especially the definition of the central metric under scrutiny) were difficult to follow or missing. In addition, the motivation for various aspects of the paper were not very clear; both the problem setting and the experiments were lacking, raising questions about the design decisions behind both.

---

> ### Author Response · Authors · 2022-08-26
> **Author Rebuttal (1/2)**
>
> > We thank the reviewer for a detailed review, and apologize if the approach was unclear in our original submission. We have updated several parts of the paper, and added additional analysis based on the reviewer's suggestion, as described below:
>
>
> First, I have concerns about the central metric being used for evaluation (or the clarity with which it is presented): the Packing Efficiency.
>
> Thank you so much for this feedback. We have updated both the naming and description of the two metrics in our work (Section 3 on Metrics, Figure 5). In summary, we rename the "packing efficiency" to be "spatial preference adherence (SPA)", and "inverse edit distance" to be "temporal preference adherence (TPA)". This renaming highlights the property each metric is trying to capture:
> (1) SPA only cares about the final state of the dishwasher, and measures the packing efficiency, while adhering to spatial preference of top or bottom rack per category. SPA is high if more objects are placed following the user's spatial preference in the dishwasher (i.e. either top or bottom rack per category).
> (2) TPA accounts for the order in which objects are picked, irrespective of where they are placed, and measures the deviation from the temporal actions of the user. TPA is high if the policy follows both sequential constraints of task (open dishwasher before loading), and the order-related preferences (bowls before cups).
> Both metrics are low if the policy repeatedly violates physical constraints of the task or does not make progress towards the task (e.g. by repeating the same action).
>
> The two metrics are meant to disentangle the two challenges of the problem we consider: spatial preferences, and temporal preferences and constraints. We hope that the updated names in Figure 5 and descriptions in Section 3.1 help alleviate the confusion caused by our original paper. Please let us know if you have further feedback on how to improve our description, and we will be more than happy to include it in the final paper.
>
> > Does this mean if any preferences are violated, the PE is zero.
>
> Yes, if all spatial preferences are violated, SPA (previously PE)  is zero. Spatial preferences are quite a strict constraint in object rearrangement, for example, users have a preferred cabinet for plates and another for cups, and it is important for a learned policy to follow this preference.
>
> The performance of the random agent is bad because we evaluate the SPA at the end of an episodic rollout. The performance of a random policy reduces exponentially with the length of the action sequence (typically around 20-30 steps long), and number of visible objects in the scene (between 6 to 20). SPA is low if the policy either (1) fails to adhere to the physical constraints, or (2) violates the spatial preference. Even if we ignore the physical constraint violations and consider the task of selecting the preferred rack for each object, the probability of a SPA of 1.0 for a random policy would be around $(1/2)^{N_objects}$, since it has to choose either top or bottom rack per object. We have updated the description of our baselines in Section 3 and added additional details about baselines in the Appendix C Metrics to clarify this point.
>
> >  Why is a bounding box used as an input for the category label rather than a one-hot encoding of the object?
>
> We have added an additional comparative experiment in the Appendix E on Discrete Category Token that compares one-hot encoding for category versus a continuous bounding box. We observe that this input choice makes the training unstable, and results in a poorer performance as compared to bounding box input. It is possible that the performance can be improved by sweeping over model parameters, and we will update final results, if so.
> Intuitively, bounding box information is useful for reasoning about 3D occupancy in the dishwasher to choose unoccupied place poses for an object, and hence can be useful for learning the physical constraints of the problem. It can also generalize to new object categories (though we do not experiment with this). For example, it might associate a knife and fork as closely related objects, and hence place them in a similar location in the dishwasher. Ideally, we would like to have more descriptive category embeddings than just bounding box, such as language description to build semantically meaningful relationships between objects.
>
> > The equation for PE seems to have an error
>
> Thank you so much for pointing this typo out, the SPA should be normalized by the number of receptacles (like the top and bottom racks). We have updated the description of the metrics in Section 3.1 on metrics. Please let us know if there are any more concerns, and we would be happy to incorporate further feedback.

---

> ### Author Response · Authors · 2022-08-26
> **Author Rebuttal (2/2) + Plots attached**
>
> > The experimental setup is not particularly well justified in the text.
>
> We apologize for any confusion caused, and address reviewer’s concerns below:
>
> * *The rate of failures associated with constraints borne of these manipulation constraints is not reported in the text*
>
> Thank you for this suggestion. We have included an additional analysis in the Appendix which evaluates the different failure causes of TTP, disentangling failures related to preference and physical constraints. Figure 16 in Appendix D shows that, in the current setting, TTP learns physical constraints of the problem perfectly, and never takes an action that violates them. Instead, the failure cases stem from either (1) not following user preferences, or (2) repeating certain actions which don't violate physical constraints, but also don't make progress towards the task, like repeatedly picking and placing the same object. In contrast, our baselines like continuous pose prediction suffer from violating task constraints, like attempting to place an object in an already occupied location. We hope that this additional analysis clarifies the rate of failure associated with manipulation constraints versus preference violations.
>
> * *Second, expecting the agent to explicitly enumerate those actions feels like an artificial limitation*
>
> We disagree with this comment. TTP is designed to learn both physical constraints of the task and user preferences from user demonstrations. In fact, one of the strengths of TTP is that it can disentangle the task constraints from preferences by learning over multiple preferences. Specifically, there are several reasons why reviewer’s comment is not applicable to the current scenario:
>
> Physical constraints are not just opening/closing racks, but also include reasoning about what location an object can fit in, given the current state of the dishwasher. This requires 3D occupancy reasoning about the currently picked object as well as other objects already in the dishwasher.
> We use a motion planner for planning robot movement, but do not assume access to a task plan. Motion planners are task-agnostic, and RRT-like sampling is enough to get good obstacle-avoidance. However, task plans are very task specific, and can be arbitrarily complex especially for complex problems like dishwasher loading. Such problem-specific constraints are tedious to specify for complex tasks, and difficult to modify for new preferences, especially for non-experts.
>
> We present an alternative to traditional Task Planning, and show that demonstrations of a task are sufficient to learn task constraints, and an architecture like TTP naturally adapts to new preferences or dynamically appearing objects. For an end-user, a prompt is an easier way to provide preference and requires no expert understanding of how to frame it in a planning language.
>
> Finally, we point the reviewer to the analysis in Figure 16 that shows that TTP learns physical constraints of the problem perfectly, and never takes an action that violates them. Since an additional task planner would not alleviate these failure cases, we don’t think that providing this additional information is necessary in our current experiments.
>
>
> > why the order in which objects are loaded into the dishwasher is a particularly salient concern
>
> Thank you for this suggestion, we have updated our paper to better motivate this point using other tasks like cooking in Section 1 - Introduction. In general, user demonstrations encode both sequential constraints of the task, like open dishwasher before loading, as well as possibly other preferences, like loading dirtier bowls before easier to clean cups. While the actual preference might be 'load as many of the dirtier bowls as possible, before worrying about easy to clean cups', it often manifests in the order in which the sub-tasks are demonstrated. For example, to maximize the number of dirtier bowls, a user might load bowls before cups. This is even more obvious in other household tasks, like cooking. For example, one user may start cooking potatoes before frying onions, because potatoes take much longer to cook, while another user might fry the onions before adding in potatoes for flavor - the order depends on their preference. In the future, we would like to experiment with other ways of encoding preferences, like language, which can be more descriptive than just order of sub-tasks.

---

### Official Review · Reviewer_999X · 2022-07-31

**Originality:** Very Good
**Technical Quality:** Excellent
**Clarity Of Presentation:** Good
**Impact:** 4

**Recommendation:**

Weak Accept: I recommend accepting the paper, but will not argue for my recommendation if the majority of other reviewers have a different opinion.

**Summary:**

This paper casts rearrangement planning as a sequence modelling problem, and applies the transformer architecture to solve it. The user prompts the model with a demonstration, e.g. placing a few plates into a dishwasher, from which the model infers the user’s preferences for arranging the scene. Given a new set of objects, the model can then predict a sequence of actions, including opening drawers and moving objects, to complete a rearrangement which is consistent with the preferences in the prompt. Target poses are selected from a set of discrete options. Experiments show that this particular transformer architecture is well-suited to modelling these rearrangement sequences, generalises well in several ways, and the method is successfully validated with a real robot demo.

**Issues:**

1. I think it needs to be made clearer early in the paper why the robot cannot simply copy the prompt actions. It is because the prompt set of objects are different to those in the situation (as explained in the video). For example, in the real world, the user might demonstrate preferences with a couple of plates, and the robot should extend this preference to the whole object set with many more plates. If the only difference between the prompt and the situation were the initial poses of the objects, then simply copying the prompt actions might be fine.
1. The related work discussion can be made more precise to the specific problem addressed by the paper. For example, instead of discussing the general benefits of machine learning approaches on Line 19, this space can be used to discuss the related work in more detail, and explain in more detail where this method extends on it (e.g. here is where we extend beyond [23]).
1. It would be useful for the reader if the paper could discuss the relationship between graph attention networks and transformers, since the former are used in prior work discussed in this paper [23,24]. When applied to this use-case, the two architectures share many similarities (computing attention between all objects in the scene). So, if prior work has already applied a similar attention mechanism, then where does the novelty come from in applying transformers? Perhaps transformers are a better fit than GATs when rearrangement planning is viewed as a sequence modelling problem?
1. This is a minor thing, but some of the formal notation can be clarified to be more consistent. Indexing starts from 0 in some places and from 1 in others when referring to the same variables. E.g. see indexing of x in lines 56 and 61. It does not affect the clarity of the overall paper too much, however.
1. Line 256: possible typo. Should this be “not overfitting” or “not underfitting”? I think not underfitting. Because if the model is overfitting, then it would still benefit from the additional training data. Whereas if it were underfitting, it would not benefit because it is struggling to fit the data it already has, if I understand this argument correctly.
1. The limitations section provides a good overview of future work, but could benefit from also having a frank discussion about inherent limitations due to the design choices in the method, for example the need to have a pre-defined set of poses to choose from.

**Quality Of The Limitations Section:**

Additional details required

**Reviewer Expertise:**

5: The reviewer is absolutely certain that the evaluation is correct and very familiar with the relevant literature

**Robotics Focus:**

Sufficient demonstration on hardware

**Strengths And Weaknesses:**

Strengths:
1. The technical quality and execution of the paper is impressive, in adapting this powerful transformer architecture for the rearrangement setting and addressing a challenging rearrangement scenario. There are many useful insights and techniques in the paper which the community will benefit from seeing used. For example, the dot-product technique for selecting an instance used on Line 132.
1. The authors have thought carefully about how they envisage the user interacting with this system (e.g. through preference prompts), which makes it a promising direction for deployment to robots working with users.
1. Unifying actions for moving objects and opening/closing containers is done elegantly, and is a useful capability for rearrangement methods to expand from tabletop environments to more complex scenes.
1. The real-world demo is useful, because it gives insights on which parts of the method make it well-suited for deployment on the real robot and which components are challenging to transfer.
1. Investigating the effect of the number of objects, especially in the out-of-distribution setting, is interesting. There is insightful analysis about failure cases, e.g. closing the dishwasher early, which is useful for future work to build on this. Other useful ablations are also performed.

Limitations:
1. A major contribution of this paper seems to be encoding the preferences of a user for the order in which objects are placed. However, the importance of this capability needs to be motivated more persuasively. If the end arrangement is the same, why does the user care about the order in which objects were moved there? Without this motivation, the impact of this contribution is not very clear. One way to frame this might be: human demonstrations contain not only preferences over end states, but clues about how to achieve those end states (e.g. open the dishwasher first). That is why encoding the order of actions is also useful.
1. Discretization of the solution space is a limitation of this method in its current form. If I understand correctly, the target pose for an object is picked from a set of pre-defined options. In simulation, these options can indeed be generated automatically. But how can these options be generated for real scenes? Does an engineer have to define a set of possible target poses for each object (the appendix suggests this)? Does a model of the environment have to be created and loaded into simulation? This also makes it difficult to apply this method to rearrangement problems without the grid-like structure of a dishwasher, for example lining up mugs on a shelf, since the number of options would need to be large to achieve precise alignment. Have the authors explored training this method to regress poses instead, i.e. make continuous pose predictions?
1. In the current version of the method, the object features focus on the bounding box and could be made more semantically expressive. I.e., can the method tell a fork and knife apart if they have the same bounding box? Does it know that they are both cutlery, and so they share semantic features and so they should be placed similarly? To be fair, this is addressed in the limitations section.
1. There are some limitations with the experimental procedure, and places where it could be described more clearly. “Packing efficiency” is used as a metric, which makes sense for dishwasher loading, but the use of this metric makes it difficult to intuitively understand how good a method is, and how good it would be on other tasks, compared to a metric such as distance error from an acceptable position for an object. Additionally, this packing efficiency is set to zero if the policy violates a preference (Line 187), which I assume includes arranging the drawers in a different order. It might be good to have an experiment which uses a metric solely measuring how good the end state is, since in most scenarios it is unclear why the user would care about anything else, so long as the end state is the same. There are some useful details about the experiment setup that are in the appendix but could go in the main paper to make it self-contained as a description of the experiments. Specifically: full examples/descriptions of what constitutes a preference in these experiments. Additionally, a more self-contained description of how the GNN-IL baseline method works would help.

**Summary Of Recommendation:**

This is a technically strong paper which addresses a challenging rearrangement problem: predicting how to load a dishwasher well. To improve it further, there are a few limitations to address which would further motivate its contributions and clarify experimental details. If these limitations can be overcome, this work will likely have a major impact on the field of object rearrangement and will be useful for the robot manipulation community at large.
- - -
EDIT: thank you to the authors for their hard work in completing the rebuttal. Taking the paper and discussion into consideration, I have decided to keep my recommendation of Weak Accept. The main reasons for this are as follows:
* The authors have argued more persuasively for the importance of temporal preferences. This is particularly evident in the cooking context. To improve this work further, it would be interesting to see experiments that show this method applied to additional tasks such as this, rather than focusing on the single task of loading a dishwasher, where preferences may more easily be inferred from the end state.
* I agree with the claim in the rebuttal which says that discrete prediction is easier that continuous prediction **if** you have a set of target poses to choose from which are all feasible. It is clear how this makes the task easier for the model. The limitation is that densely sampling these target poses is challenging for real-world scenes. One way forward, as the authors suggest, is a pre-defined dictionary of suitable poses based on the specific model of the dishwasher that the user has in their home. However, this does make the method less easy to apply and limits the tasks for which it can be used. That is why this is a limitation of the method. However, loading items into dishwashers, drying racks, ovens, etc is still useful for a robot, and so the paper is sufficiently impactful. I encourage the authors to briefly mention this hurdle to real-world deployment in the limitations section of the final main paper, since they do agree with this and mention it in the appendix. The rebuttal mentions that this was included in Section 6, but the limitations section still seems to be Section 5 in the updated paper, and does not seem to mention this point, if I am reading it correctly.
* Thank you to the authors for their clarifications on the experimental setup, and the detailed responses in general. Much appreciated!

Overall, the paper is technically well-executed and addresses a useful task in home robotics, so it will make a valuable addition to CoRL 2022.

---

> ### Author Response · Authors · 2022-08-26
> **Author Rebuttal (1/2)**
>
> We thank the reviewer for the detailed, insightful comments. We are encouraged to see that the reviewer found “technical quality and execution of the paper is impressive”, “user interacting with this system (e.g. through preference prompts)” well thought out, “Unifying actions” elegant, “real-world demo is useful” and “analysis about failure cases” insightful .
> We have addressed several raised concerns below:
>
> > "If the end arrangement is the same, why does the user care about the order in which objects were moved there?"
>
> This is a great question, and a central premise of our work, so thank you for pointing this out. In general, user demonstrations encode both sequential constraints of the task, like open dishwasher before loading, as well as possibly other preferences, like loading dirtier bowls before easier to clean cups. While the actual preference might be 'load as many of the dirtier bowls as possible, before worrying about easy to clean cups', it often manifests in the order in which the sub-tasks are demonstrated. For example, to maximize the number of dirtier bowls, a user might load bowls before cups. This is even more obvious in other household tasks, like cooking. For example, a user may start cooking potatoes before frying onions, because potatoes take much longer to cook, while another might fry the onions for flavor before adding in potatoes - the order depends on their preference. Our approach, TTP, is able to disentangle the sequential nature of task constraints from preferences, and learns both, while generalizing to new preferences at test time. In the future, we would like to experiment with other ways of encoding preferences, like language, which can be more descriptive than just order of sub-tasks. We have updated our paper to better explain this point using other tasks than just dishwasher loading in Section 1 introduction.
>
> > Discretization of the place solution space is a limitation of this method in its current form.
>
> Thank you for this insightful comment. We have conducted an additional experiment with continuous place pose prediction for our task, and added it to Appendix E on Continuous Placement Pose. We observe that continuous space prediction performs poorly (SPA of 0.05) compared to discrete pose selection on our problem (SPA of 0.62). The main failure case over discrete pose is predicting infeasible poses for objects. This is probably because of the geometry of a dishwasher, where small prediction errors can make a target place position infeasible. For example, a plate can only fit in a particular pose, and small orientation errors make the placing infeasible. This makes dense, overlapping, discrete poses more suitable for this environment. In the real-world, one could imagine creating a dense place pose dictionary based on the model of the dishwasher in the user's home. However, for more general receptacles, it is possible that continuous prediction might be more suitable, and an approach like our continuous baseline, might be better suited.
>
> Note that while we use discrete poses, they are densely sampled, and not mutually exclusive. Hence, several discrete poses overlap, and if one pose is occupied by an object, there might be several others that are also occupied by the same object. TTP has to reason about the size of an object and distances between place poses to decide which pose is empty to place the next object, while maximizing packing efficiency. Hence, our policy is able to reason about 3D space occupancy, even if working with discretized poses.
>
> > The object features could be made more semantically expressive. To be fair, this is addressed in the limitations section.
>
> We agree with the reviewer that ideally category embeddings should be more descriptive than just a bounding box, and include some language labels like knife versus fork. In our current work, we assumed that similar sized objects have similar occupancy footprint, and this was enough to reason about feasible placing locations in a rack. This was conducive to learning physical constraints of objects in the scene, but ignores other relationships between objects. In future work, we would like to explore additional multi-modal object features, including language descriptions along with the geometry information.
>
> > There are some limitations with the experimental procedure, and places where it could be described more clearly.
>
> Thank you for raising these points, we have made several changes to our manuscript based on your suggestions, as detailed below.

---

> ### Author Response · Authors · 2022-08-26
> **Author Rebuttal (2/2) + Plots attached**
>
> > relationship between graph attention networks and transformers, since the former are used in prior work discussed in this paper ... Perhaps transformers are a better fit than GATs when rearrangement planning is viewed as a sequence modeling problem?
>
> Thank you for this insightful comment. Indeed, GATs and Transformers are closely related network architectures. In theory, a GAT architecture over a fully-connected graph with an additional time-based feature would be equivalent to our Transformer-based model. We are certainly not the first to apply attention-based architectures to planning problems, but unlike [7,8], we consider sequential planning over multiple timesteps, which is critical to understanding human preferences [a]. We have highlighted these differences in Appendix C.
>
> Another difference from GNN-based prior work on sequential planning [7, 8] is that we use cross attention to condition the policy on the prompt, instead of adding preferences as an input feature. To do this, we use a slot attention architecture to learn a preference representation, and re-use instance encoders across prompt and situation. While these architectural choices are possible in GNNs, they are more intuitive and faster to implement in Transformers. A full step-by-step comparison of the architectural and training changes is currently out of scope, but we will try to  add this to the appendix of final paper to help in the broader discussion of GNNs versus Transformers for the community.
>
> > “Packing efficiency” is used as a metric, ... the end state is.
>
> Thank you for this suggestion. We have updated the name and explanation of "packing efficiency" to "spatial preference adherence (SPA)" in the paper. SPA only cares about the final state of the dishwasher, and maximizes placed objects, while adhering to spatial preference. SPA is high if more objects are placed following the user's spatial preference in the dishwasher (i.e. either top or bottom rack per category).
>
> Specifying the exact desired location of every object is difficult for a user, especially in settings where the number of objects is variable, and a spatial goal is more natural. For example, people usually have one cabinet for cups, while another for plates, instead of exact desired locations for all plates and cups. Hence, we believe that for generalized object rearrangement spatial goals are more natural and scalable than specifying positions per object.
>
> > There are some useful details about the experiment setup that are in the appendix but could go in the main paper to make it self-contained as a description of the experiments.
>
> Thank you, we will update our paper and bring this description back to the paper in Section 3.
>
> > Additionally, a more self-contained description of how the GNN-IL baseline method works would help.
>
> We have added a new section and figure in the Appendix C and Figure 12 which explains the GNN based baselines in detail.
>
>
> > I think it needs to be made clearer early in the paper why the robot cannot simply copy the prompt actions
>
> Thank you for pointing this out, you are absolutely correct. The policy cannot simply copy the prompt actions because the prompt object set is different from the situation. We have updated our paper to better highlight this point (Section 2.1 on Prompt-Situation Transformer).
>
> > The related work discussion can be made more precise to the specific problem addressed by the paper.
>
> Thank you for this suggestion, and we will update the final paper to highlight the differences from prior related work Section 4 better.
>
> > Typos, minor errors and limitations
>
> Again, thank you for your detailed review! We have updated the errors that you pointed out in our paper. We have also updated the limitation Section 6 to talk about the choice of discrete place poses.
>
> ---
> ### References
>
> *[a] Mandlekar, Ajay, et al. "What matters in learning from offline human demonstrations for robot manipulation." arXiv preprint arXiv:2108.03298 (2021).*
>
> *[7] Y. Lin, A. S. Wang, E. Undersander, and A. Rai. Efficient and interpretable robot manipulation with graph neural networks. IEEE Robotics and Automation Letters, 2022.*
>
> *[8] ​​I. Kapelyukh and E. Johns. My house, my rules: Learning tidying preferences with graph neural networks. In Conference on Robot Learning, pages 740–749. PMLR, 2022.*

---

### Author Response · Authors · 2022-08-27
**Summary and Updated paper**

We thank all the reviewers and the meta reviewer for the valuable comments and suggestions that helped us improve our paper. We are encouraged that the reviewers found the work is well motivated (Reviewer 3v7A, Reviewer qr8n) and practically-grounded (Reviewer 999X).

- Technical contribution (Reviewer 999X , Reviewer 3v7A), experimental in-depth analysis (Reviewer 999X , Reviewer 3v7A, Reviewer qr8n)

- Interesting and challenging dishwasher-loading application including real world demo (Reviewer 999X , Reviewer 3v7A, Reviewer qr8n) with unified actions (Reviewer 999X) and dynamically appearing objects (Reviewer qr8n)

- Clear writing (Reviewer 3v7A, Reviewer qr8n), comprehensive literature survey (Reviewer 3v7A, Reviewer qr8n)

- Breaks the barrier between two fields - Robotics and NLP (Reviewer 3v7A)

Below we address the common points raised by the reviewers, and highlighted in the meta review. In summary, we have changed the following aspects of our paper:

- Clarified the metrics and renamed them to better capture what they are measuring (Section 3).

- Analysis of the errors incurred by TTP in simulation (Appendix D and Figure 16).

- Additional comparison experiments against continuous place pose prediction and categorical distribution for object categories (Figure 20).

- Detailed description of baselines in appendix (C.1). Additional baseline experiment where preference embeddings are learned for the GNN-IL baselines (C.4).

---

### Meta-Review · Area_Chair_VbzS · 2022-08-15

**Recommendation:** Accept (Poster)
**Confidence:** 4

**Metareview:**

Below is a summary of the strengths and weaknesses of the paper, according to the reviewers.

Strengths:

- The method proposed is well motivated and has several interesting technical contributions.
- The example application of loading a dishwasher helps to show intuition behind the method, and shows the practicality of the method.
- There is a real-world demo.
- The ablation studies are particularly useful.

Weaknesses:

- It is not clear why the order in which items are placed, is an important concept to encode in a user's preferences.
- Discretisation of the solution space limits the practicality of the method in real-world settings.
- There are concerns in the motivation for, and the clarity in the description of, the "packing efficiency" metric.
- There are concerns about the experimental setup regarding failures associated with physical constraints.

In the rebuttal, please address the above weaknesses, as well as the other concerns and questions raised by the reviewers.

---------

Update after the rebuttal:

On average, the reviewers increased their recommendation following the rebuttal, with all four reviewers now recommending acceptance. This is a technically strong paper tackling a challenging problem, with a useful real-world demo addressing a classic dishwasher loading task, which many in the community will be able to relate to. However, reviewers found the motivation for some aspects of the method to be a little unclear, so these should be clarified in the final paper.

**Best Paper Nomination:**

No

---

> ### Author Response · Authors · 2022-08-26
> **Author Rebuttal (1/2)**
>
> We thank the reviewers for valuable feedback. We are encouraged that the reviewers found the work is well motivated ([Reviewer 3v7A](), [Reviewer qr8n](https://openreview.net/forum?id=Eal_lL08v_l&noteId=IRlu8onj3mi)) and practically-grounded ([Reviewer 999X](https://openreview.net/forum?id=Eal_lL08v_l&noteId=dOoBlpszivk)).
>
> - Technical contribution (Reviewer 999X , Reviewer 3v7A), experimental in-depth analysis (Reviewer 999X , Reviewer 3v7A, Reviewer qr8n)
>
> - Interesting and challenging dishwasher-loading application including real world demo (Reviewer 999X , Reviewer 3v7A, Reviewer qr8n) with unified actions (Reviewer 999X) and dynamically appearing objects (Reviewer qr8n)
>
> - Clear writing (Reviewer 3v7A, Reviewer qr8n), comprehensive literature survey (Reviewer 3v7A, Reviewer qr8n)
>
> - Breaks the barrier between two fields - Robotics and NLP (Reviewer 3v7A)
>
> > “It is not clear why the order in which items are placed is an important concept to encode in a user's preferences.”
>
> This is a great question, and a central premise of our work, so thank you for pointing this out. In general, user demonstrations encode both sequential constraints of the task, like open dishwasher before loading, as well as possibly other preferences, like loading dirtier bowls before easier to clean cups. While the actual preference might be 'load as many of the dirtier bowls as possible, before worrying about easy to clean cups', it often manifests in the order in which the sub-tasks are demonstrated. For example, to maximize the number of dirtier bowls, a user might load bowls before cups. This is even more obvious in other household tasks, like cooking. For example, *a user may start cooking potatoes before frying onions, because potatoes take much longer to cook, while another might fry the onions for flavor before adding in potatoes - the order depends on their preference*. Our approach, TTP, is able to disentangle the sequential nature of task constraints from preferences, and learns both, while generalizing to new preferences at test time. In the future, we would like to experiment with other ways of encoding preferences, like language, which can be more descriptive than just order of sub-tasks. We have updated our paper to better explain this point using other tasks than just dishwasher loading in Section 1 introduction.
>
> > “Discretisation of the solution space limits the practicality of the method in real-world settings.”
>
> This is not entirely correct. We consider two solution spaces: decision over ‘what to pick’, and ‘where to place’. The pick solution space is naturally discretized by the objects of interest visible in the environment (dishwasher, racks, dishes, etc.), and does not limit real-world transfer. Note that while we consider discrete objects, their features like location are continuous. An object segmentation pipeline, as used in our real-world experiments, can provide the discrete set of objects to interact with, and their continuous features.
>
> On the other hand, our choice of discretized 'where to place’ solution space can be seen as a hurdle to real-world transfer. Our motivation for this choice was twofold:
> (1) In our experiments, dense discrete poses generalized to unseen scenarios better than continuous prediction. We provide additional experiments verifying this observation in Appendix E on Continuous Placement Pose.
> (2) The geometry of a dishwasher is well-suited for discrete positions, as small prediction errors can make a target place position infeasible. For example, a plate can only fit in a particular pose, and small orientation errors make the placing infeasible. In the real-world, one could imagine creating a dense place pose dictionary based on the model of the dishwasher in the user's home. However, we agree that for other receptacles, like a shelf, this discretization is not necessary, and a continuous prediction approach like our baseline, might be better suited. We have discussed this point better in the main paper, and added additional comparison experiments against continuous predictions to Appendix E on Continuous Placement Pose.
>
> Figure 20 shows that discrete placement pose instances achieve higher SPA than continuous place pose prediction, while the TPA is comparable to original TTP. This is because TTP-ContP policy learns to pick objects in the correct order like TTP but suffers in placement pose prediction.
>
> Finally, we note that while we use discrete place poses, they are densely sampled, and not mutually exclusive. Hence, several discrete poses overlap, and if one pose is occupied by an object, there might be several others that are also occupied by the same object. TTP has to reason about the size of an object and distances between place poses to decide which pose is empty to place the next object, while maximizing packing efficiency. Hence, our policy is able to reason about 3D space occupancy, even if working with discretized poses.

---

> ### Author Response · Authors · 2022-08-26
> **Author Rebuttal (2/2) + Plots attached**
>
> > There are concerns in the motivation for, and the clarity in the description of, the "packing efficiency" metric.
>
> Thank you for this question, and we apologize for the confusion caused by our original description. We have clarified the metrics in our paper better by updating their names and descriptions (Section 3 on Metrics, Figure 5). In summary, we rename the "packing efficiency" to be "spatial preference adherence (SPA)", and "inverse edit distance" to be "temporal preference adherence (TPA)". This renaming highlights the property each metric is trying to capture:
> (1) SPA only cares about the final state of the dishwasher, and measures the packing efficiency, while adhering to spatial preference of top or bottom rack per category. SPA is high if more objects are placed following the user's spatial preference in the dishwasher (i.e. either top or bottom rack per category).
> (2) TPA accounts for the order in which objects are picked, irrespective of where they are placed, and measures the deviation from the temporal actions of the user. TPA is high if the policy follows both sequential constraints of task (open dishwasher before loading), and the order-related preferences (bowls before cups).
> Both metrics are low if the policy repeatedly violates physical constraints of the task or does not make progress towards the task (e.g. by repeating the same action).
>
> > There are concerns about the experimental setup regarding failures associated with physical constraints.
>
> We have added an additional analysis which measures the causes of failure in our experiments (Appendix D on Failure Analysis). First, we would like to note that TTP is meant to learn both physical constraints and preferences. In fact, one of the strengths of TTP is that it can disentangle the task constraints from preferences by learning over multiple preferences. Next, we point to the analysis in Figure 16 that shows that in our experiments TTP learns physical constraints of the problem perfectly, and never takes an action that violates them. Instead, the failure cases stem from either (1) not following user preferences, or (2) repeating certain actions which don't make progress towards the task, like repeatedly picking and placing the same object. In contrast, our baselines like continuous pose prediction suffer from violating task constraints, like attempting to place an object in an already occupied location.
>
> Both (1) and (2) stem out of the problem of sequential decision making, and are some of the challenges of this problem. We hope that this analysis answers reviewers' concerns about our experimental setup. Future work should try to learn policies that can adapt from interaction and correct for such errors.